

# Atmospheric particulate matter characterization by Fourier Transform Infrared spectroscopy: a review of statistical calibration strategies for carbonaceous aerosol quantification in US measurement networks

Satoshi Takahama[1], Ann M. Dillner[2], Andrew T. Weakley[2], Matteo Reggente[1], Charlotte Bürki[1], Mária Lbadaoui-Darvas[1], Bruno Debus[2], Adele Kuzmiakova[1,7], and Anthony S. Wexler[2,3,4,5,6]

[1]ENAC/IIE Swiss Federal Institute of Technology Lausanne (EPFL), Lausanne, Switzerland
[2]Air Quality Research Center, University of California Davis, Davis, CA 95616, United States
[3]Center for Health and the Environment, University of California, Davis, CA 95616, United States
[4]Mechanical and Aeronautical Engineering, University of California, Davis, CA 95616, United States
[5]Civil and Environmental Engineering, University of California, Davis, CA 95616, United States
[6]Land, Air and Water Resources, University of California, Davis, CA 95616, United States
[7]now at Stanford University

**Correspondence:** S. Takahama (satoshi.takahama@epfl.ch)

**Abstract.**

Atmospheric particulate matter (PM) is a complex mixture of many different substances, and requires a suite of instruments for chemical characterization. Fourier Transform Infrared (FT-IR) spectroscopy is a technique that can provide quantification of multiple species provided that accurate calibration models can be constructed to interpret the acquired spectra. In this capacity,

FT-IR has enjoyed a long history in monitoring gas-phase constituents in the atmosphere and in stack emissions. However, application to PM poses a different set of challenges as the condensed-phase spectrum has broad, overlapping absorption peaks and contributions of scattering to the mid-infrared spectrum. Past approaches have used laboratory standards to build calibration models for prediction of inorganic substances or organic functional groups and predicting their concentration in atmospheric PM mixtures by extrapolation.

In this work, we review recent studies pursuing an alternate strategy, which is to build statistical calibration models for mid-IR spectra of PM using collocated ambient measurements. Focusing on calibrations with organic carbon (OC) and elemental carbon (EC) reported from thermal optical reflectance (TOR), this synthesis serves to consolidate our knowledge for extending FT-IR to provide TOR-equivalent OC and EC measurements to new PM samples when TOR measurements are not available. We summarize methods for model specification, calibration sample selection, and model evaluation for these substances at

several sites in two US national monitoring networks: 7 sites in the Interagency Monitoring of PROtected Visual Environments (IMPROVE) network for the year 2011, and 10 sites in the Chemical Speciation Network (CSN) for the year 2013. We then describe application of the model in an operational context for the IMPROVE network for samples collected in 2013 at 6 of the same sites as 2011, and 11 additional sites. In addition to extending the evaluation to samples from a different year and different sites, we describe strategies for error anticipation due to precision and biases from the calibration model to assess




model applicability for new spectra a priori. We conclude with a discussion regarding past work and future strategies for recalibration. In addition to targeting numerical accuracy, we encourage model interpretation to facilitate understanding of the underlying structural composition related to operationally-defined quantities of TOR OC and EC from the vibrational modes in mid-IR deemed most informative for calibration. The paper is structured such that the life cycle of a statistical calibration

model for FT-IR can be envisioned for any substance with IR-active vibrational modes, and more generally for instruments requiring ambient calibrations.

## Contents



*Copyright statement.*  TEXT

## 5    1    Introduction

Airborne particles are made of inorganic salts, organic compounds, mineral dust, black carbon, trace elements, and water (Seinfeld and Pandis, 2006). While regulatory limits on airborne particulate matter (PM) concentrations are set by gravimetric mass determination, analysis of chemical composition is desired as it provides insight into source contributions, facilitates evaluation of chemical simulations, and strengthens links between particle constituents and health and environmental impacts.

However, the diversity of molecular constituents pose challenges for characterization as no single instrument can measure all relevant properties; an amalgam of analytical techniques are often required for comprehensive measurement (Hallquist et al., 2009; Kulkarni et al., 2011; Pratt and Prather, 2012; Nozière et al., 2015; Laskin et al., 2018). Fourier Transform Infrared (FT-IR) spectroscopy is one analytical technique that captures the signature of a multitude of PM constituents that give rise to feature-rich spectral patterns over the mid-infrared (mid-IR) wavelengths (Griffiths and Haseth, 2007). In the past decade, mid-

IR spectra have been used for quantification of various substances in atmospheric PM, and for apportionment of organic matter (OM) into source classes including biomass burning, biogenic aerosol, fossil fuel combustion, and marine aerosol (Russell et al., 2011). The quantitative information regarding the abundance of substances in each spectrum is limited only by the calibration models that can be built for it.

In principle, the extent of frequency-dependent absorption in the mid-IR accompanying induced changes in the dipole
moment of molecular bonds can be used to estimate the quantity of sample constituents in any medium (Griffiths and Haseth, 2007). Based on this principle, FT-IR has a long history in remote and ground-based measurement of chemical composition in the atmospheric vapor phase (Griffith and Jamie, 2006). For ground-based measurement, gases are measured by FT-IR in an open-path in-situ configuration (Russwurm and Childers, 2006), or via extractive sampling into a closed, multi-pass cell (Spellicy and Webb, 2006). These techniques have been used to sample urban smog (Pitts et al., 1977; Tuazon et al.,
1981; Hanst et al., 1982); smog chamber (Akimoto et al., 1980; Pitts et al., 1984; Ofner, 2011), biomass burning emissions (Hurst et al., 1994; Yokelson et al., 1997; Christian et al., 2004), volcanoes (Oppenheimer and Kyle, 2008), and fugitive gases (Kirchgessner et al., 1993; Russwurm, 1999; U.S. EPA, 1998); emission fluxes (Galle et al., 1994; Griffith and Galle, 2000; Griffith et al., 2002), greenhouse gases (Shao and Griffiths, 2010; Hammer et al., 2013; Schütze et al., 2013; Hase et al., 2015); and isotopic composition (Meier and Notholt, 1996; Flores et al., 2017). For these applications, quantitative analysis has been
conducted using various regression algorithms with standard gases or synthetic calibration spectra with absolute accuracies on



the order of 1–5%. Synthetic spectra for calibration are generated from a database of absorption line parameters together with simulation of pressure and Doppler broadening, and instrumental effects (Griffith, 1996; Flores et al., 2013).

Analysis of FT-IR spectra of condensed-phase systems are more challenging. PM can be found in crystalline solid, amorphous solid, liquid, and semi-solid phase states (Virtanen et al., 2010; Koop et al., 2011; Li et al., 2017). Solid and liquid-phase

spectra do not have the same rotational lineshapes present in the vapor phase, but inhomogeneous broadening occurs due to a multitude of local interactions of bonds within the liquid or solid environment (Turrell, 2006; Griffiths and Haseth, 2007; Kelley, 2013). Lineshapes are particularly broad in complex mixtures of atmospheric PM, since the resulting spectrum is the superposition of varying resonances for a given type of bond. FT-IR has enjoyed a long history of qualitative analysis of molecular characteristics in multicomponent PM based on visible peaks in the spectrum (e.g., Mader et al., 1952; Presto et al., 2005;

Kidd et al., 2014; Chen et al., 2016a), and study of relative composition or changes to composition under controlled conditions (e.g., humidification, oxidation) has provided insight into atmospherically-relevant aerosol processes (e.g., Cziczo et al., 1997; Gibson et al., 2006; Hung et al., 2013; Zeng et al., 2013). Quantitative prediction of substances in collected PM presents a separate task, and is conventionally pursued by generating laboratory standards and relating observed features to known concentrations. This calibration approach has been predominantly used to characterize ambient and atmospherically-relevant

particles collected on filters or optical disks. The bulk of past work in aerosol studies have focused on using laboratory standards to build semi-empirical calibration models for individual vibrational modes belonging to one of many functional groups present in the mixture. In this approach, the observed absorption is related to a reference measurement (typically gravimetric mass) of the compounds on the substrate. In this way, calibration of nitrate and sulfate salts (Cunningham et al., 1974; Cunningham and Johnson, 1976; Bogard et al., 1982; McClenny et al., 1985; Krost and McClenny, 1992, 1994; Pollard et al., 1990;

Tsai and Kuo, 2006; Reff et al., 2007), silica dust (Foster and Walker, 1984; Weakley et al., 2014; Wei et al., 2017), and organic functional groups (Allen and Palen, 1989; Paulson et al., 1990; Pickle et al., 1990; Mylonas et al., 1991; Palen et al., 1992, 1993; Holes et al., 1997; Blando et al., 1998; Maria et al., 2002, 2003; Sax et al., 2005; Gilardoni et al., 2007; Reff et al., 2007; Coury and Dillner, 2008; Day et al., 2010; Takahama et al., 2013; Faber et al., 2017) have been studied. The organic carbon and organic aerosol mass reconstructed has typically ranged between 70–100% when compared with collocated evolved-gas

analysis or mass spectrometry measurements (Russell et al., 2009; Corrigan et al., 2013), though many model uncertainties remain. One is that unmeasured, non-functionalized skeletal carbon can lead to less than full mass recovery, and the second is the estimation of the detectable fraction due to the multiplicity of carbon atoms associated with each type of functional group. (Maria et al., 2003; Takahama and Ruggeri, 2017). The challenge in this type of calibration is in the problem of extrapolating from the reference composition, which is necessarily kept simple, to that of the chemically complex PM. Spectroscopically,

this difference can lead to shifts in absorption intensity or peak locations, and a general broadening of absorption peaks on account of the same functional group appearing in many different molecules and in different condensed-phase environments.

Synthetic spectra for condensed-phase systems can be generated by mechanistic and statistical means, but are not readily available for quantitative calibration. Absolute intensities are typically even more difficult to simulate accurately for than peak frequencies (Gussoni et al., 2006). Computational models that predict vibrational motion of molecules in isolation using

quantum mechanical models (Barone et al., 2012) or by harmonic approximation for larger molecules (Weymuth et al., 2012)





suffer from two shortcomings: poor treatment of anharmonicity and lack of solvent effects in liquid solutions (Thomas et al., 2013). Quantum mechanical simulations can parameterize interactions with an implicitly modeled solvent through a polariz-able continuum model framework (Cappelli and Biczysko, 2011), but do not adequately represent specific interactions such as hydrogen bonding (Barone et al., 2014). Microsolvation can be a better technique to describe hydrogen bonding environment

but the high computational cost prevents application to large systems (Kulkarni et al., 2009). Gaussian dispersian analysis has provided accurate spectrum reconstruction in pure liquids (water-ethanol mixtures) from their calculated dielectric functions (MacDonald and Bureau, 2003), but has not been applied to more complex systems. Molecular dynamics (MD) provides a general framework for addressing interactions with the solvent, large-amplitude motions in flexible molecules, and anhar-monicities (Ishiyama and Morita, 2011; Ivanov et al., 2013). Electronic structure calculations relevant for predicting vibrational

spectra can be incorporated by ab initio MD (Car and Parrinello, 1985; Marx, 2009; Thomas et al., 2013), and path integral MD methods such as centroid or ring polymer MD (Witt et al., 2009; Ceriotti et al., 2016) that additionally considers nuclear quantum effects (at higher computational cost). Ab initio MD is widely used for the simulating the spectra of water and a range small organic and biological molecules in isolation (Silvestrelli et al., 1997; Aida and Dupuis, 2003; Gaigeot et al., 2007; Gaigeot, 2008; Thomas et al., 2013; Fischer et al., 2016) Such calculations generally reproduce the shape of the spectrum well

with respect to experimental ones at very high dilution, although C-H stretching peaks are known to be shifted towards higher wavenumbers due to the lack of improper hydrogen bonding in vacuum simulations (Thomas et al., 2013). Bulk liquid phase simulations are limited to a few tens of molecules (few hundreds of atoms), and have been performed for liquids, including methanol (Thomas et al., 2013), water (Silvestrelli et al., 1997), and aqueous solutions of biomolecules (Gaigeot and Sprik, 2003). These simulations reproduce peak positions and relative intensities sufficiently well when compared to experimental

spectra, albeit with lower accuracy in peak position at wavenumbers higher than $2000 \ \mathrm{cm}^{-1}$. These methods have also been shown to reproduce main features of vibrational spectra in solid (crystalline ice and naphthalene) systems (Bernasconi et al., 1998; Putrino and Parrinello, 2002; Pagliai et al., 2008; Rossi et al., 2014b). Nuclear quantum effects not explicitly accounted for by ab initio calculations become more important for hydrogen-containing systems, and have been investigated in liquid water and methane for vibrational spectra simulation (Rossi et al., 2014a, b; Medders and Paesani; Marsalek and Markland,

2017). While such methods may be useful in aiding interpretation of environmental spectra (Kubicki and Mueller, 2010; Pe-done et al., 2010), they are not yet mature for reproducing spectra of suitable quality for quantitative calibration or (white-box) inverse modeling.

Early applications of artificial intelligence to mid-IR spectra interpretation also included efforts to generate synthetic spec-tra of individual compounds. Mid-IR spectra of new compounds were simulated from neural networks trained on three-

dimensional molecular descriptors (radial distribution functions) paired with corresponding mid-IR spectra, matched by simi-larity (nearest neighbor) search in a structural database, or generated from substructure/spectral correlation databases (Dubois et al., 1990; Weigel and Herges, 1996; Baumann and Clerc, 1997; Schuur and Gasteiger, 1997; Selzer et al., 2000; Yao et al., 2001; Gasteiger, 2006). Drawing upon internal or commercial libraries (Barth, 1993), predictions were made for compounds in the condensed phase with a diverse set of substructures including including methanol, amino acids, ring-structured acids,

and substituted benzene derivatives. Many structural features including peak location, relative peak heights, and peak widths




were reproduced, provided that relevant training samples were available in the library. Much of the work was motivated by pattern matching and classification of spectra for unknown samples (Robb and Munk, 1990; Novic and Zupan, 1995), and automated band assignment and identification of the underlying fragments typically performed by trained spectroscopists (Sasaki et al., 1968; Gribov and Elyashberg, 1970; Christie and Munk, 1988; Munk, 1998; Hemmer, 2007; Elyashberg et al., 2009).

This approach has been able to generate spectra for more complex molecules than mechanistic modeling relying on ab initio calculations. However, the extent of evaluation has been limited; extension to multicomponent mixtures and usefulness for quantitative calibration is currently not known. While these research fields remain an active part of cheminformatics, we propose another approach for calibration model development that can be used for atmospheric PM analysis.

As an alternative to laboratory-generated mixtures and simulated spectra, collocated measurements of substances for which

there are IR-active vibrational modes can be used as reference values for calibration. This data-driven approach permits the complexity of atmospheric PM spectra with overlapping absorbances from both analytes and interferences to be included in a calibration model. For instance, Allen et al. (1994) demonstrated the use of collocated ammonium sulfate by ion chromatography to quantify the abundance of this substance from FT-IR spectra, though some uncertainties arose from the time resolution between the sampling instruments. Reference measurements of vibrational mode or functional group composition

in atmospheric PM are typically not available, though other (often requiring more sample mass and user-labor) measurement techniques such as ultraviolet-visible spectrometry or nuclear magnetic resonance spectroscopy (Decesari et al., 2003; Ranney and Ziemann, 2016) for quantification may be considered for comparison (subject to their higher minimum detection limits). For this paper, we focus on calibration of FT-IR spectra with collocated measurements for prediction of carbonaceous aerosol content. Organic carbon (OC) and elemental carbon (EC) can be characterized by evolved gas analysis (EGA, which includes

thermal optical reflectance, or TOR; and thermal optical transmittance, or TOT) and used for calibration of FT-IR spectra. EGA OC and EC are widely-measured in monitoring networks (Chow et al., 2007; Brown et al., 2017), with historical significance in regulatory monitoring, source apportionment, and epidemiological studies. While EC is formally defined as sp2-bonded carbon bonded only to other carbon atoms, what is measured by EGA EC is an operationally-defined quantity which is likely associated with low-volatility organic compounds (Chow et al., 2004; Petzold et al., 2013; Lack et al., 2014). EGA OC comprises a

larger fraction of the total carbon and therefore less influenced by pyrolysis artifacts that affects quantification of EGA EC. In addition to OC estimates independently constructed from laboratory calibrations of functional groups, prediction of EGA OC and EC from FTIR spectra will provide values for which strong precedent in atmospheric studies exist.

The benefit of building data-driven calibration models to reproduce reported concentrations by measurements already available is twofold. One is to enable FT-IR to serve as an alternative measurement technique when the standard suite of instruments

are not available. For example, FT-IR spectra can be acquired rapidly, non-destructively, and at low cost from from Polytetrafluoroethylene (PTFE) filters commonly used for gravimetric mass analysis in regulatory monitoring and epidemiological studies. This capability is beneficial when a single filter may be used for many types of analysis during short term campaigns, or in network sites for which installation of the full suite of instruments is prohibitive (e.g., EGA requires a separate sampler for quartz fiber filters). The second benefit is the ability to gain a better understanding of atmospheric constituents measured by other




techniques (e.g., operationally defined quantities such as EGA OC and EC) by associating them with important vibrational modes structural elements of molecules identified in the FT-IR calibration model.

In this paper, we review the current state-of-the art for quantitative prediction of OC and EC as reported by TOR using FT-IR in selected sites of the Interagency Monitoring of PROtected Visual Environments (IMPROVE) monitoring network and
Chemical Speciation Network (CSN). We place this work within the context of overseeing the life cycle of a statistical calibration model more generally; reporting further developments in anticipating errors due to precision and bias in new samples, and describing a roadmap for future work. While partial least squares regression and its variants figure heavily in the calibration approach taken thus far, related developments in the fields of machine learning, chemometrics, and statistical process monitoring are mentioned to indicate the range of possibilities yet available to overcome future challenges in interpreting complex the
mid-IR spectra of PM. We expect that many concepts described here will also be relevant for the emerging field of statistical calibration and deployment of measurements in a broader environmental and atmospheric context (e.g., Cross et al., 2017; Kim et al., 2017; Zimmerman et al., 2018). In the following sections, we describe the experimental methods for collecting data (Section 2), the calibration process (Section 3), assessing suitability of existing models for new samples (Section 4.1), and maintaining calibration models (Section 4.2). Finally, we conclude with a summary and outlook (Section 5).

## 2  Background

First, we review the basic principles of FT-IR and how the measured absorbances can be related to underlying constituents, including carbonaceous species (Section 2.1). We then describe the samples used for calibration and evaluation (Section 2.2). We then conclude the section with discussion regarding quality assurance and quality control (QA/QC) of the FT-IR hardware performance (Section 2.3). Under the assumption that these hardware QA/QC criteria are met, we dedicate the remainder of
the paper outlining model evaluation on the assumption that the performance in prediction can be attributed to differences in sample composition.

### 2.1  Fourier Transform-Infrared Spectroscopy

In this section, we cover the background necessary to understand FT-IR spectroscopy in the analysis of PM collected onto filter media (PTFE). The wavelengths of IR are longer than visible light (400-800 nm) and FT-IR refers to a non-dispersive
analytical technique probing the mid-IR, which is radiation from 2,500 nm to 25,000 nm or in the vibrational frequency units used by spectroscopists, wavenumbers, 4000 to 400 $\mathrm{cm}^{-1}$. Molecular bonds absorb mid-IR at characteristic frequencies of their vibrational modes when interactions between electric dipole and electric field induce transitions among vibrational energy states (Steele, 2006; Griffiths and Haseth, 2007). Based on this principle, the spectrum obtained by FT-IR represents the underlying composition of organic and inorganic functional groups containing molecular bonds with a dipole moment.
In transmission-mode analysis where the IR beam is directed through the sample, absorbance ($A$) can be obtained by ratioing the measured extinction of radiation through the sample ($I$) by a reference value ($I_0$), also called the "background", and taking



the negative value of their decadic logarithm (first relation of eq. 1).

$$A(\tilde{\nu}) = -\log_{10}\left[\frac{I(\tilde{\nu})}{I_0(\tilde{\nu})}\right] = \epsilon(\tilde{\nu})n^{(a)} \tag{1}$$

The sample is the PTFE filter (with or without PM) and the background is taken as the empty sample compartment. The quality of the absorbance spectrum depends on how accurately the background reflects the conditions of the sample scan, and the
background is therefore acquired regularly as discussed in Section 2.3.

When absorption is the dominant mode of extinction, the measured absorbance ($A$) is proportional to the areal density of molecules ($n^{(a)}$) in the beam in the sample (eq. 1) (Duyckaerts, 1959; Kortüm, 1969; Nordlund, 2011). $\epsilon$ is the proportionality constant and is called the molar absorption coefficient. Although scattering off of surfaces present in the sample can generate a significant contribution to the absorbance spectrum, its effects can be modeled as a sum of incremental absorbances by a linear
calibration model, or minimized through spectral pre-processing procedures (baseline correction) as discussed in Section 3.3.1.

A composite metric of PM such as carbon content presumably results from contributions by a myriad of substances. The abundances of these underlying molecules concurrently give rise to the apparent mass of carbon ($m_C$) (eq. 2) measured by evolved gas analysis and the absorbance spectrum ($A$) (eq. 3) measured by FT-IR (Ottaway et al., 2012):

$$m_C^{(a)} = 12.01 \cdot \sum_k f_{C,k} n_k^{(a)} \tag{2}$$

$$A(\tilde{\nu}) = \sum_k \epsilon_k(\tilde{\nu})\, n_k^{(a)} + \sum_{k'} \epsilon_{k'}(\tilde{\nu})\, n_{k'}^{(a)} + \{\ldots\} \ . \tag{3}$$

$f_{C,k}$ denotes the number of (organic or elemental) carbon in molecule $k$, and 12.01 is the atomic mass of carbon. Non-carbonaceous substances (e.g., inorganic compounds) that give rise to additional (possibly interfering) absorbance are indexed by $k'$. The superscript "(a)" denotes an area-normalized quantity. "$\{\ldots\}$" indicates contributions from instrumental noise, ambient background, and additional factors such as scattering. Using TOR measurements from collocated quartz fiber filters, our
objective is to develop a calibration model for estimating the abundance of carbonaceous material ($m_C^{(a)}$) in the PTFE sample that may have led to the observed pattern of mid-IR absorbances ($A(\tilde{\nu})$). A common approach is to explore the relationship between response and absorbance spectra through a class of models which take on a multivariate, linear form (Griffiths and Haseth, 2007):

$$m_{C,i}^{(a)} = \sum_j b_j A_i(\tilde{\nu}_j) + e_i \ . \tag{4}$$

The set of wavelength-dependent regression coefficients $b_j$ comprise a vector operator that effectively extracts the necessary information from the spectrum for calibration. These coefficients ($b_j$s) presumably represent a weighted combination of coefficients expressed in eqs. 2 and 3 (also correcting for non-carbonaceous interferences). The remaining term, $e_i$, characterizes the model residual (in regression fitting) or prediction error (in application to new samples). The relationship with underlying substances ($k$) that comprise OC and EC is implicit, though some efforts to interpret these constituents have been made through
examination of latent (or hidden) variables obtained from the calibration model (discussed in Section 3.4).





Using complex, operationally-defined TOR measurements as reference for calibration, some caution in interpretation and application is warranted. For instance, these coefficients may not necessarily capture the true relationship expressed by eqs. 2 and 3, but rather rely on correlated rather than causal variables for quantification. Particles and the PTFE substrate itself can confer a large scattering contribution to the extinction spectrum (eq. 1), and additional sample matrix interactions among

analytes may challenge assumptions regarding the linear relationship (eq. 3) underlying the model for quantification (eq. 4) (Geladi and Kowalski, 1986). Furthermore, the relationship between spectra and concentrations embodied by $b$ is specific to the the chemical composition of PM at the geographic location and sampling artifacts due to composition and sample handling protocols of the calibration samples.To address these concerns, extensive evaluation regarding model performance in various extrapolation contexts are necessary to investigate the limits of our calibration models, and methods for anticipating prediction

errors provide some guidance on their general applicability in new domains. Regression coefficients and underlying model parameters are inspected to determine important vibrational modes that provide insight into the infrared absorption bands that drive the predictive capability of our regression models.

## 2.2 Sample collection (IMPROVE and CSN)

The IMPROVE network consists of approximately 170 sites in rural and pristine locations in the United States primarily

National Parks and Wilderness Areas. Data from the IMPROVE network is used to monitor trends in particulate matter concentrations and visibility. IMPROVE collects ambient samples midnight to midnight every third day by pulling air at 22.8 liters per minute through filters. Polytetrafluoroethylene (PTFE, 25 mm, Pall Corp.) or more commonly referred to as Teflon filters are routinely used for gravimetric, elemental, and light absorption measurements and are used in this work for FT-IR analysis. Quartz filters are used for thermal optical reflectance (TOR) measurements to obtain organic and elemental carbon. Nylon

filters are used to measure inorganic ions, primarily sulfate and nitrate.

The CSN consists of about 140 sites located in urban and suburban area and the data is used to evaluate trends and sources of particulate matter. Ambient samples are collected in the CSN on midnight to midnight schedule one every third or one every sixth day. Quartz filters for TOR analysis are collected with a flowrate of 22.8 lpm. PTFE filters (Whatmam $PM_{2.5}$ membranes, 47 mm, used through late 2015; MTL filters (47 mm) have been used there after) and nylon filters are collected at a flowrate of

6.7 lpm flowrate. All sites in CSN have used TOR analysis for carbon analysis since 2010.

TOR analysis consists of heating a portion of the quartz filter with the IMPROVE_A temperature ramp and measuring the evolved carbon (Chow et al., 2007). The initial heating is performed with an inert environment and the material that is removed is ascribed to organic carbon (OC). Oxygen is added at the higher temperatures and the measured material is ascribed to elemental carbon (EC). Charring of ambient particulate carbon is corrected using a laser that reflects off the surface of

the sample (hence reflectance) (Chow et al., 1993). The evolved carbon is converted to methane and measured with a flame ionization detector. Organic carbon data is corrected for gas-phase adsorption using a monthly median blank value specific to each network ("Change to artifact correction method for OC carbon fractions").

For this work, we examine a subset of these sites in which PTFE filters were analyzed for FT-IR spectra (Figure 1). For model building and evaluation (Section 3), we use 7 sites consisting of 794 samples for IMPROVE in 2011, and 10 sites



consisting of 1035 samples for CSN in 2013. Two sites in IMPROVE 2011 are samplers collocated at the same urban location in Phoenix, AZ, and one site (Sac and Fox) that was discontinued mid-year. Additional IMPROVE samples were analyzed by FT-IR during sample year 2013, which included 6 of the same sites and 11 additional sites. This data set is used for evaluation of the operational phase of the model (Section 4).

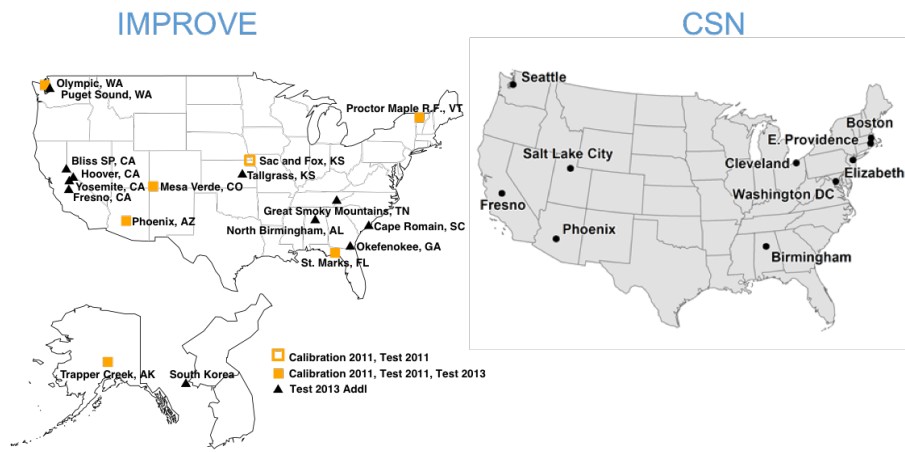

**Figure 1.** Map of IMPROVE and CSN sites used for this work. The Sac and Fox, KS, IMPROVE site was only operational for the first half of 2011.

## 2.3 Laboratory operations and quality control of analysis

IMPROVE and CSN PTFE sample and blank filters are analyzed without pretreatment on either Tensor 27 or Tensor II FT-IR instruments (Bruker Optics, Billerica, MA) equipped with a liquid nitrogen-cooled detector. Filters are placed in a small, custom-built sample chamber which reliably places each filter the same distance from the source. IR-active water vapor and $CO_2$ are purged from the sample compartment and instrument optics to minimize absorption bands of gas phase compounds in the aerosol spectra. Samples are measured in transmission mode and absorbance spectra, which are used for calibration and prediction, are calculated using the most recent empty chamber spectrum as a reference (collected hourly). The total measurement time for one filter is 5 minutes. Additional details on the FT-IR analysis are described by Ruthenburg et al. (2014) and Debus et al. (2018).

Daily and weekly quality control checks are performed to monitor the comparability, precision and stability of the FT-IR instruments. Duplicate spectra are collected every fifty filters (once or twice per day) per instrument in order to evaluate measurement precision. Measured precision values are low and smaller than the 95th percentile of the standard deviation of the blanks for both TOR OC and EC indicating that instrument error has a relatively minor influence on the prediction of TOR OC and EC and is smaller than the variability observed between PTFE filters. Quality control filters — blank filters and ambient samples — are analyzed weekly to monitor instrument stability. Debus et al. (2018) conclude that predictions of TOR OC and EC remain relatively stable over a two and a half year period based on analyses of quality control filters,





and that observed changes are small. These data enable us to track instrumental changes that will require re-calibration. A subset of ambient filters are analyzed on all FT-IR instruments to evaluate spectral dissimilarities and differences in prediction. These samples show that differences in spectral response between instruments are small and due mainly to variability in PTFE. In addition, these samples indicate that careful control of laboratory conditions and detector temperature, sample position,

relative humidity (RH) and $CO_2$ levels in the FT-IR instrument enables instrument-agnostic calibrations that predict accurate concentrations independent of the instrument on which a spectrum is collected. The quality control data show that the TOR OC and EC measurements obtained from multiple FT-IR instruments in one laboratory are precise, stable (over the 2.5 year period evaluated) and agnostic to instrument used for analysis (Debus et al., 2018).

## 3    Model building, evaluation, and interpretation

In this section, we describe the model building process for quantitative calibration. The relationship betwen spectra and reference values to be exploited for prediction can be discovered using any number of algorithms, method of spectra pretreatment, and the *calibration set* of samples to be used for model training and validation. As the best choices for each of these categories are not known a priori, the typical strategy is to generate a large set of candidate models and select one that scores well across a suite of performance criteria against a *test set* of samples reserved for independent evaluation. The process of building and

evaluating a model conceptualized in the framework of statistical process control is depicted in Figure 2. In the first stage, various pathways to model construction are evaluated, and expectations for model performance are determined. The second stage involves continued application and monitoring of model suitability for new samples (*prediction set*), which is discussed in Section 4.1. Where applicable, the sample type in each data set should include several types of samples. For instance, the calibration set can include blank samples in which analyte (but not necessarily interferent) concentrations are absent. Test and

prediction set samples can include both analytical and field blank samples. Collocated measurements can be used for providing replicates for calibration, or used as separate evaluation of precision. Immediately below, we describe the procedure for model specification, algorithms for parameter estimation, and model selection in Section 3.1. Methods for spectra processing are described in Section 3.3, and sample selection in Section 3.5. In each section, the broader concept will be introduced and then its application to TOR will be reviewed.

## 25    3.1    Model estimation

Many algorithms in the domain of statistical learning, machine learning, and chemometrics have demonstrated utility in building calibration models with spectra measurements: neural networks (Long et al., 1990; Walczak and Massart, 2000), Gaussian process regression (Chen et al., 2007), support vector regression (Thissen et al., 2004; Balabin and Smirnov, 2011), principal components regression (Hasegawa, 2006), ridge regression (Hoerl and Kennard, 1970; Tikhonov and Arsenin, 1977; Kalivas,

2012), wavelet regression (Brown et al., 2001; Zhao et al., 2012), functional regression (Saeys et al., 2008), partial least squares (Rosipal and Krämer, 2006); among others. There is no lack of algorithms for supervised learning with continuous response variables that can potentially be adapted for such an application (Hastie et al., 2009). Each of these techniques map relation-





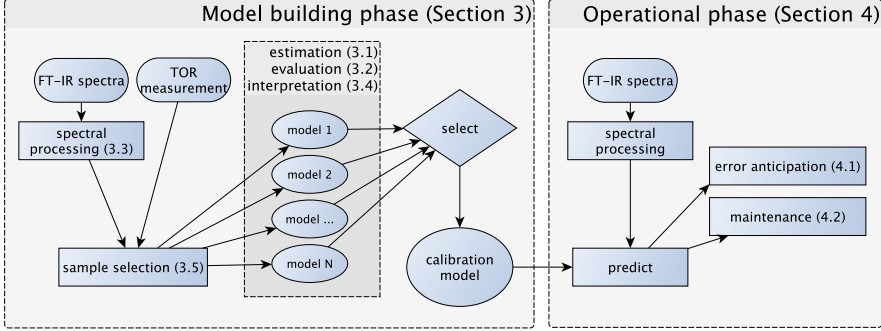

**Figure 2.** Diagram of the model building, evaluation, and monitoring process. Sections and subsections covering the illustrated topics are denoted in parentheses. Note that the any of the calibrations $\{1, 2, \ldots, N\}$ can be a multilevel model (Section 3.5.3) consisting of an ensemble of models.

ships between spectral features and reference concentrations using different similarity measures, manifolds, and projections; largely in metric spaces where the notion of distances among real-valued data points are well-defined (e.g., Zezula et al., 2006; Russolillo, 2012). The best mathematical representation for any new data set is difficult to ascertain a priori, but models can be compared by their fundamental assumptions and their formulation: e.g., linear or non-linear in form; globally parametric,

locally parametric, or distribution free (random forest, nearest neighbor); feature transformations; objective function and constraints; and expected residual distributions. Approaches that incorporate randomized sampling can return slightly different numerical results, but reproducibility of any particular result can be ensured by providing seed values for the pseudo-random number generator. A typical procedure for model development is to select candidate methods that have enjoyed success in similar applications and empirically investigate which techniques provide meaningful performance and interpretability for the

current task, after which implementation measures are then pursued (Kuhn and Johnson, 2013). In lieu of selecting a single model, ensemble learning and Bayesian model averaging approaches combine predictions from multiple models (Murphy, 2012).

For FT-IR calibration targeting prediction of TOR-equivalent concentrations, we focus on finding solutions to the linear model introduced in Section 2.1. Letting $\boldsymbol{y} = [m_{C,i}/a]$, $\boldsymbol{X} = [A_i(\tilde{\nu}_j)]$, $\boldsymbol{b} = [b_i]$, and $\boldsymbol{e} = [e_i]$, we re-express equation 4 in array

notation to facilitate further discussions of linear operations:

$$\boldsymbol{y} = \boldsymbol{X}\boldsymbol{b} + \boldsymbol{e} \, . \tag{5}$$

Equation 5 is an ill-posed inverse problem; therefore, it is desirable to introduce some form of regularization (method of introducing additional information or assumptions) to find suitable candidates for $\boldsymbol{b}$ (Zhou et al., 2005; Friedman et al., 2010; Takahama et al., 2016). In this paper, we summarize the application of partial least squares (PLS) (Wold, 1966; Wold et al.,

2001) for obtaining solutions to this equation, with which good results have been obtained for our application and FT-IR spectra more generally (Hasegawa, 2006; Griffiths and Haseth, 2007). This technique has been a classic workhorse of chemometrics for many decades and is particularly well suited for characteristics of FT-IR analysis, for which data are collinear (neighboring




absorbances are often related to one another) and high-dimensional (more variables than measurements in many scenarios). These issues are addressed by projection of spectra onto an orthogonal bases of latent variables (LVs) that take a combination of spectral features, and regularization by LV selection (Andries and Kalivas, 2013). Furthermore, PLS is agnostic with respect to assumption of residual structure (e.g., normality) for obtaining $b$, which circumvents the need to explicitly account for

covariance or error distribution models to characterize the residuals (Aitken, 1934; Nelder and Wedderburn, 1972; Kariya and Kurata, 2004). PLS is also used as a preliminary dimension reduction technique prior to application of non-linear methods (Walczak and Wegscheider, 1993) (an approach for outlier detection is described in Section 4.1.2). Therefore, it is sensible that PLS should be selected as a canonical approach for solving eq. 5.

Mathematically, classical PLS represents a bilinear decomposition of a multivariate model in which both $X$ and $y$ are

projected onto basis sets ("loadings") $P$ and $q$, respectively (Wold et al., 1983, 1984; Geladi and Kowalski, 1986; Mevik and Wehrens, 2007):

$$X = TP^T + E_X$$
$$y = Tq^T + e \ .$$

(6)

$T$ is the orthogonal score matrix and $E_X$ denotes the residuals in the reconstruction of the spectra matrix. Common solution methods search for a set of loading weight vectors (represented in a column matrix $W$) such that covariance of scores ($T$)

with respect to the response variable ($y$) is maximized. The weight matrix can be viewed as a linear operator that changes the basis between the feature space and FT-IR measurement space. These weights and their relationship to the score matrix and regression vector are expressed below:

$$R = W \left( P^T W \right)^{-1}$$
$$T = XR$$
$$b = Rq^T \ .$$

(7)

For univariate $y$ as written in equation 5, a number of commonly used algorithms — Nonlinear Iterative PArtial Least Squares

(NIPALS; Wold et al., 1983), SIMPLS (deJong, 1993), kernel PLS (with linear kernel; Lindgren et al., 1993) — can be used to arrive at the same solution (while varying in numerical efficiency). Kernel PLS can be further extended into modeling non-linear interactions by projecting the spectra onto a high-dimensional space and applying linear algebraic operations akin to classical PLS, with comparative performance to support vector regression and other commonly-used non-linear modeling approaches (Rosipal and Krämer, 2006). However, likely due to the linear nature of the underlying relationship 4, linear PLS has typically

performed better than non-linear algorithms for FT-IR calibration (Griffiths and Haseth, 2007). In addition, the linearity of classical PLS regression has yielded more interpretable models than non-linear ones (Luinge et al., 1995). Therefore, past applications of PLS to FT-IR calibration of atmospheric aerosol constituents has focused on its linear variants and will be the focus of this paper.

An optimal number of LVs must be selected to arrive at the best predictive model. A larger number of LVs is increasingly

able to capture the variations in the spectra, leading to reduction in model bias. Some of the finer variations in the spectra are





not part of the analyte signal which we wish to model; including LVs that model these terms lead to increased variance in its predictions. A universal problem in statistical modeling is to find a method for characterizing model bias and variance such that one with the lowest apparent error can be chosen. There is no shortage of methods devised to capture this bias-variance tradeoff and their implications for model selection continue to be an active area of development (Hastie et al., 2009). With no
immediate consensus on the single best approach for all cases, the approach often taken is to select and use one based on prior experience until found to be inadequate (as with model specification).

One class of methods characterize the bias and variance using the information obtained from fitting of the data. For instance, Akaike Information Criterion (AIC; Akaike, 1974) and Bayesian Information Criterion (BIC; Schwarz, 1978) consider the balance between model fidelity (fitting error, which monotonically decreases with number of parameters) with penalties incurred
for increasing model complexity (which serves as a form of regularization). The fitting error may be characterized by residual sum of squares or maximum likelihood estimate (e.g., Li et al., 2002), and the penalty may be a scaled form of the number of parameters or norms of the regression coefficient vector. An effective degrees of freedom (EDF) or generalized EDF parameter aims to characterize the resolvable dimensionality as apparent from the model fit to data (Tibshirani, 2014), though the EDF may not always correspond to desired model complexity (Krämer and Sugiyama, 2011; Janson et al., 2015).

Another class of methods relies on assessment of the bias and variance contributions implicitly present in prediction errors, which are obtained by application of regression coefficients estimated using a training data set and evaluated against a separate set of ("validation") data withheld from model construction to fix its parameters. To maximize the data available for both training and validation, modern statistical algorithms such as cross validation (Mosteller and Tukey, 1968; Stone, 1974; Geisser, 1975) and the bootstrap method (Efron and Tibshirani, 1997) allows use of the same samples for both training and validation,
which comprise what we collectively refer to as the calibration set. The essential principle is to partition the same calibration set multiple times such that the model is trained and then validated on different samples over a repeated number of trials. In this way, a distribution of performance metrics for models containing different subsets of the data can be aggregated to determine a suitable estimate of a parameter (number of LVs). The number and arrangement of partitions vary by method, with cross-validation using each sample exactly once for validation and bootstrap resamples with replacement. Both have reported usable
results (Molinaro et al., 2005; Arlot and Celisse, 2010). For increasingly smaller number of samples, Leave-One-Out (LOO) CV or bootstrap may be favored as it reserves a larger number of samples to train each model, though it is generally appreciated that LOO leads to suboptimal estimates of prediction error (Hastie et al., 2009). Evaluation metrics are calculated on samples which have not been involved in the model-building process (Esbensen and Geladi, 2010). Examples of metrics include the minimum root-mean-square error of cross validation (RMSECV) (one of the most widely used metrics; Gowen et al., 2011),
one standard deviation above RMSECV (Hastie et al., 2009), Wold's $R$ criterion (Wold, 1978), coefficient of determination ($R^2$), randomization $p$-value (van der Voet, 1994; Wiklund et al., 2007), among others. A suite of these metrics can also be considered simultaneously (Zhao et al., 2015). The final model is obtained by refitting the model to all of the available samples in the calibration set and using the number of parameters selected in the CV process. Other strategies and general discussions on the topic of performance metrics and statistical sampling are covered in many textbooks (e.g., Bishop, 2009; Hastie et al.,
2009; Kuhn and Johnson, 2013).





Past work on TOR and FT-IR measurements have used $V$-fold CV, with Dillner and Takahama (2015a; 2015b) using minimum RMSECV and Weakley et al. (2016) using Wold's $R$ criterion for performance evaluation. In $V$-fold CV, the data is partitioned into $V$ groups, and $V$-1 subsets are used to train a model to be evaluated on the remaining subset (repeated for $V$ arrangements). Dillner and Takahama (2015a) found that $V$=2, 5, and 10 selected different number of LVs but led to similar

overall performance. To keep the solution deterministic (i.e., no random sampling) and representative (i.e., the composition of training sets and validation sets are representative of the overall calibration sets across permutations), samples in the calibration set are ordered according to a strategy amenable for stratification. For instance, samples are arranged by sampling site and date (used as a surrogate for source emissions, atmospheric processing, and composition, which often vary by geography and season), or with respect to increasing target analyte concentration, and samples separated by interval $V$ are used to create

each partition in a method referred to as Venetian blinds (also referred to as interleaved or striped) CV. An illustration of RMSECV compared to the fitting errors represented by the root-mean-square error calibration (RMSEC) for TOR OC is shown in Figure 3. Other strategies for arranging CV include maximizing differences among samples in each fold to reduce chances of overfitting (Kuhn and Johnson, 2013) but has not been explored in this application.

Even with specification of model and approach for parameter selection fixed, spectral processing and sample selection can

lead to differences in overall model performance. We first discuss how different models can be generated from the same set of samples according to these decisions before proceeding to protocols for model evaluation using the *test set* reserved for independent assessment (Section 3.2). The test set is used to compare the merits of models built in different ways, and establish control limits for the operational phase (Section 4).

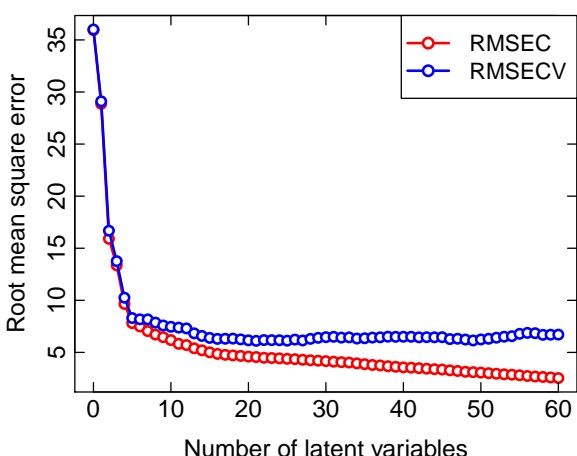

**Figure 3.** Illustration of RMSEC, which represents the fitting errors, and RMSECV, which represents the prediction error, calculated for TOR OC using the same calibration set. 10-fold Venetian blinds CV was used for this calculation.





## 3.2 Model evaluation

Statistical models can be evaluated using many of the same techniques also used by mechanistic models (Olivieri, 2015; Seinfeld and Pandis, 2016). In this section, we describe methods for evaluating overall performance (Section 3.2.1) and occurrence of systematic errors (Section 3.2.2).

### 3.2.1 Overall performance

Predictions for the set of selected models for 2011 IMPROVE and 2013 CSN data sets are shown in Figure 4. There are many aspects of each model which we wish to evaluate by comparing predictions against known reference values. These aspects include the bias and magnitude of dispersion, but also our capability to distinguish ambient samples from blank samples at the low end of observed concentrations. Metrics which capture these effects can effectively be derived from the term $e$ in the multivariate regression equation (eq. 5) when predictions and observations are compared in the test set spectra. $e$ is referred to as the residual when describing deviations from observations in fitted values, and prediction error when describing deviations from observed values when the model is used for prediction in new samples. However, by convention we often resort to the negative of the residual such that deviation in prediction is calculated with respect to the observation, rather than the other way around. Example distributions for residuals and prediction errors for TOR OC in 2011 IMPROVE are shown in Figure 5.

While the use of the minimum root-mean-square error (RMSE) is pervasive in chemometrics and machine learning as a formal parameter tuning or model selection criterion, another family of metrics are more commonly used in the air quality community (Table 1). For instance, the mean bias and mean error and their normalized quantities are often used for model-measurement evaluation of mechanistic (chemical transport) models (Seinfeld and Pandis, 2016). $R^2$ is commonly used in inter-comparisons of analytical techniques. Many of the statistical estimators in Table 1 converge to a known distribution from which confidence intervals can be calculated; or otherwise estimated numerically (e.g., by bootstrap). In addition to conventional metrics, alternatives drawing upon robust statistics (Huber and Ronchetti, 2009) are also useful when undue influence from a few extreme values may to lead to misrepresentation of overall model performance (Barnett and Lewis, 1994). For instance, the mean bias is replaced by the median bias, and mean absolute error is replaced by median absolute deviation. Even if a robust estimator is unbiased, it may not have the same variance properties as its non-robust counterpart (Venables and Ripley, 2003); therefore, comparison against a reference distribution for statistical inference may be less straightforward.

For TOR-equivalent values predicted by FT-IR, the median bias and errors have been typically preferred for characterizing overall model performance, together with $R^2$ and the minimum detection limit (MDL). Mean errors have been examined primarily to make specific comparisons among models. Having derived these metrics, we place them in context by comparing them to those reported by the reference (TOR) measurement, which include collocated measurement precision and percent of samples below MDL (Table 2).





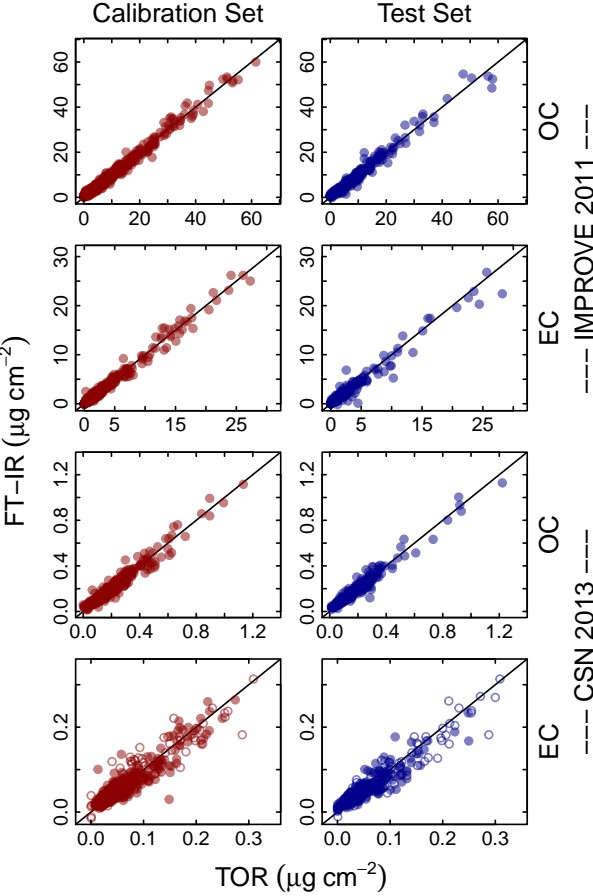

**Figure 4.** Illustration of model fits ("Calibration Set", left column) and predictions ("Test Set", right column) for the 2011 IMPROVE and 2013 CSN networks. Open circles for CSN EC indicate anomalous samples (discussed in Section 3.5.3). Note units are in areal mass density on the filter.

### 3.2.2 Systematic errors

In addition to the aggregate metrics discussed above, we evaluate whether essential effects appear to be accounted for in the regression by examining errors across different classes of samples. Systematic patterns or lack of randomness can be evaluated by examining the independence of the individual prediction errors with respect to composition, or using time and location 5 of sample collection as surrogates for composition. For instance, high prediction errors elevated over multiple days may be associated with aerosols of a particular composition transported under synoptic scale meteorology. Special exception is made for concentration, as errors can be heteroscedastic (i.e., non-constant variance) on account of the wide concentration range of atmospheric concentrations that may be addressed by a single calibration model. This heteroscedasticity leads to a distribution





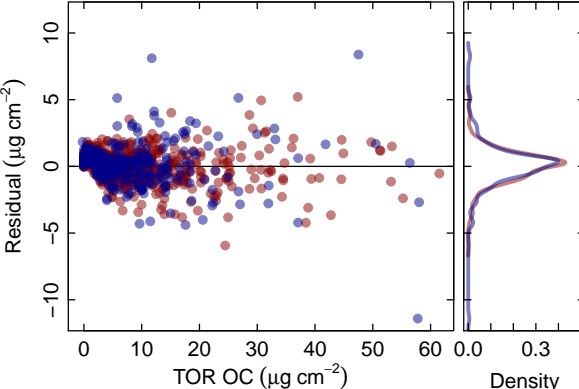

**Figure 5.** Residuals (red symbols) and prediction errors (blue symbols) from 2011 IMPROVE OC (baseline corrected, base case) predictions. The corresponding kernel density estimate of the distribution is shown on right.

**Table 1.** Definition for figures of merit for overall assessment of prediction error, samples to which they are applied, and their reference distribution (if available) used for significance testing. $\boldsymbol{y}$ is the (mean-centered) response vector (i.e., TOR OC or EC mass loadings), and $\hat{\boldsymbol{y}}$ is the predicted response (eq. 5). $\langle \cdot \rangle$ is the sample mean, Med$[\cdot]$ is the sample median, and Var$[\cdot]$ is the unbiased sample variance. $N_c$ is the number of paired collocated samples.

| Metric | Samples | Estimate | Ref. dist. |
|---|---|---|---|
| root mean square error (RMSE) | all | $\sqrt{\langle (\hat{\boldsymbol{y}} - \boldsymbol{y})^2 \rangle}$ | $\chi^2$ |
| mean bias | all | $\langle \hat{\boldsymbol{y}} - \boldsymbol{y} \rangle$ | |
| median bias | all | Med$[\hat{\boldsymbol{y}} - \boldsymbol{y}]$ | |
| mean absolute error | all | $\langle |\hat{\boldsymbol{y}} - \boldsymbol{y}| \rangle$ | $t$ |
| median absolute deviation | all | Med$[|(\hat{\boldsymbol{y}} - \boldsymbol{y}) - \text{Med}[\hat{\boldsymbol{y}} - \boldsymbol{y}]|]$ | |
| coefficient of determination ($R^2$) | all | $1 - (\hat{\boldsymbol{y}} - \boldsymbol{y})^T (\hat{\boldsymbol{y}} - \boldsymbol{y}) / (\boldsymbol{y}^T \boldsymbol{y})$ | $F$ |
| minimum detection limit (MDL) | blank | $3\sqrt{\text{Var}[(\hat{\boldsymbol{y}} - \boldsymbol{y})]}$ | $\chi^2$ |
| collocated precision | collocated | $\|\hat{\boldsymbol{y}}_1 - \hat{\boldsymbol{y}}_2\| / \sqrt{2N_c}$ | $t$ |

that is leptokurtic (i.e., heavy-tailed) compared to a normal distribution, as shown in Figure 5. As solution algorithms for PLS are agnostic with respect to such residual structure, their application to this type of problem is well-suited.

Given the propensity of prediction error distributions to be long-tailed, error and residual values are transformed to standard-normal variates using inverse hyperbolic sine (IHS) functions (Johnson, 1949; Burbidge et al., 1988; Tsai et al., 2017) using

5   parameters derived from samples with similar analyte (TOR) concentrations. Such a transformation aids identification of systematic errors in prediction related to sample collection time and location; a control chart is displayed for TOR-equivalent OC in Figure 6. Each prediction error is then characterized by its $Z$-score, which gives an immediate indication of its relation to other prediction errors for samples with similar concentrations. Because of the IHS transformation, the magnitude of errors





**Table 2.** Description and figures of merit for "base case" models. "Predictors" describe the number of wavenumbers and "Components" describe the number of LVs. Bias and errors are estimated by ensemble medians.

| Network | FT-IR | Baseline correction | Wavenumber selection | Predictors | Components |
|---|---|---|---|---|---|
| IMPROVE 2011 | OC | Spline | None | 1563 | 15 |
| | EC | Raw | None | 2784 | multilevel |
| CSN 2013 | OC | 2nd derivative | BMCUVE | 375 | 3 |
| | EC | Spline | BMCUVE | multilevel | multilevel |

| Network | FT-IR | $R^2$ | Bias ($\mu g\,m^{-3}$) | Error ($\mu g\,m^{-3}$) | MDL ($\mu g\,m^{-3}$) | Below MDL (%) | Precision ($\mu g\,m^{-3}$) |
|---|---|---|---|---|---|---|---|
| IMPROVE 2011 | OC | 0.97 | 0.01 | 0.08 | 0.11 | 0.7 | 0.21 |
| | EC | 0.96 | 0.00 | 0.03 | 0.01 | 2 | 0.06 |
| CSN 2013 | OC | 0.95 | 0.04 | 0.15 | 0.49 | 3.0 | 0.19 |
| | EC | 0.88 | 0.02 | 0.11 | 0.17 | 4.8 | 0.04 |

| Network | TOR | MDL ($\mu g\,m^{-3}$) | below MDL (%) | Precision ($\mu g\,m^{-3}$) |
|---|---|---|---|---|
| IMPROVE 2011 | OC | 0.05 | 1.5 | 0.14 |
| | EC | 0.01 | 3 | 0.11 |
| CSN 2013 | OC | 0.51 | 2.7 | 0.23 |
| | EC | 0.03 | 16.7 | 0.09 |

do not scale linearly in vertical distance on the chart, but conveys its centrality, sign, and bounds of the error (e.g., 3 units from the mean encompasses 99% of errors in samples similar in concentration). In this data set, we can see that prediction errors for Sac and Fox (SAFOX) in each concentration regime are biased positively during the winter, but systematically trend toward the mean toward the summer months. Other high error samples near the 99th percentile ($\pm 3$ probits) occur in the urban

5  environment of Phoenix, where the TOR OC concentrations are also highest. However, the prevalence of higher errors in only one of the two Phoenix measurements (PHOE5) may be indicative of sampler differences, rather than unusual atmospheric composition. Errors are negatively biased during the summer months in Trapper Creek, when TOR OC concentrations are typically low.

Systematic errors arising from under-representation of concentration or composition range in the calibration set of IM-

10 PROVE was investigated by deliberate permutations of calibration and test set samples by Dillner and Takahama (2015a; 2015b). This study is discussed together with model interpretation (Section 3.5.1). Weakley et al. (2018a) a found systematic errors with respect to OC/EC ratios when predicting TOR-equivalent EC concentrations in the CSN network. These samples were found to originate from Elizabeth, NJ, which differed from the nine other examined sites on account of the high con-





tributions from diesel PM and extent of reduced charring compared to other samples. The solution was to build a separate calibration model (Section 3.5.3).

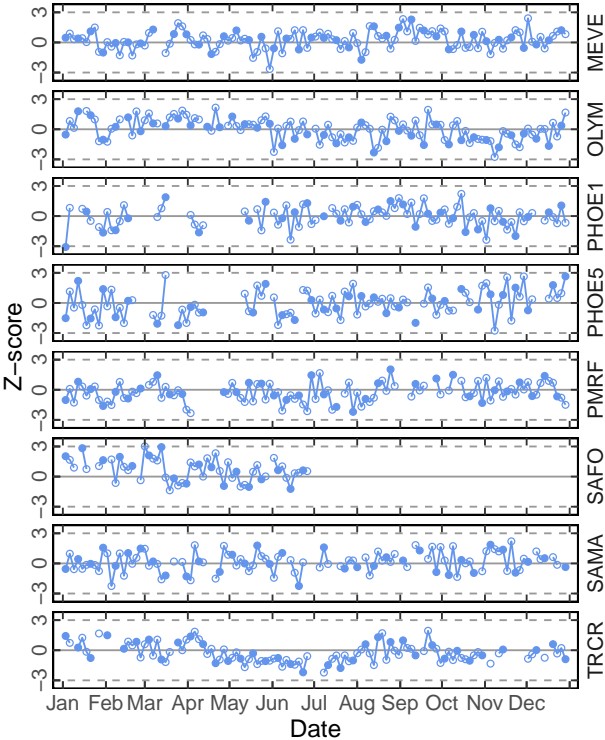

**Figure 6.** Time series chart of TOR-equivalent OC residuals (for calibration samples) and prediction errors (for test set samples) separated by site. Each value (residual: open circle, prediction error: filled circle) is mapped to a median-centered inverse hyperbolic sine function using 175 values (approximately 20% of the 2011 IMPROVE set) from neighboring TOR OC concentrations to derive distribution parameters so that values are defined within a normal distribution ($p$-value $> 0.2$). Dotted horizontal lines indicate $\pm 3$ standard deviations of the standard normal variate ($Z$-score).

## 3.3 Spectral preparation

Mid-IR spectra can be processed in many different ways for use in calibration. The primary reasons for spectral processing are to remove influences from scattering such that calibrations models follow the principles of the linear relation outlined in equation 4, and to remove unnecessary wavenumbers or spectral regions that degrade prediction quality or interpretability. Scattering of particles manifests itself in a broad contribution to the signal that is present in the measured spectrum by FT-IR and is addressed by a class of statistical methods referred to as baseline correction (Section 3.3.1). It is even possible to model nonlinear relationships such as the scattering contribution to the signal by a linear model with additional LVs, but these phenomena may not be mixed together with the noise (Borggaard and Thodberg, 1992; Despagne and Luc Massart,



1998). Elimination of unnecessary wavenumbers can reduce noise in the predictions and confer interpretation on the important absorption bands used for prediction; the class of procedures used in this is referred to as variable selection, uninformative variable elimination, among other names (Section 3.3.2). Some algorithms can separate the influence of the background and select variables in the process of finding the optimal set of coefficients $b$ in eq. 5. In each of the following sections, the each of

the topics in spectral processing will be introduced before describing their applications to TOR calibrations.

### 3.3.1   Baseline correction

Baseline correction can be fundamental to the way spectra are analyzed quantitatively. Significant challenges exist in separating the analyte signal from the baseline of mid-IR spectra, which include the superposition of broad analyte absorption bands (O-H stretches in particular) to the broadly varying background contributions from scattering. The algorithm for baseline correction

may therefore depend on the type of analyte and the broadness of its profile; optimization of the correction becomes more important as concentrations decrease such that they become difficult to distinguish from the baseline. Approaches can be categorized as reference-dependent or reference-independent (Rinnan et al., 2009), and can be handled within or outside of the regression step. Reference-dependent methods define the baseline with respect to an external measurement, which may be a reference spectrum (Afseth and Kohler, 2012) or concentrations of an analyte. For instance, orthogonal signal correction (OSC)

(Wold et al., 1998) isolates contributions to the spectrum that are uncorrelated with the analyte, and can be conceptualized as containing baseline effects. OSC can be incorporated into PLS in which the orthogonal contribution would be represented by underlying LVs (Trygg, 2002). Even without explicit specification of orthogonal components, the influence of baseline effects is accounted for by multiple LVs in the standard PLS model (Dillner and Takahama, 2015a). Reference-independent baseline correction methods remove baseline contributions based on the structure of the signal without invocation of reference values.

Two examples described below include interpolation and derivative correction methods. A more comprehensive discussion on this topic is provided by Rinnan et al. (2009).

While theories for absorption peak profiles are abundant, the lack of corollaries for baselines (Dodd and DeNoyer, 2006) lead to semi-empirical approaches for modeling their effects. If we conceptualize the broad baseline as an $N$-th order polynomial, we can approximate this expression with an analytical function or algorithm. Models can be considered to be (globally) parameteric

(e.g., polynomial, exponential) across a defined region of a spectrum, or non-parameteric (e.g., spline or convex hull; Eilers, 2004) in which case local features of the spectrum are considered with more importance. These approaches typically determine the form of the curve by training a model on regions without significant analyte absorption, and interpolated through the analyte region. The modeled baseline is then subtracted from the raw spectrum such that the analyte contribution remains. Model parameters are selected such that processed spectra conform to physical expectations — namely, that blank absorbances are

close to zero and analyte absorbances are non-negative. In general, these approaches aim to isolate the absorption contribution to the spectra that are visually recognizable, and therefore most closely conform to traditional approaches for manual baseline removal used by spectroscopists. In addition to quantitative calibration or factor analytic applications (e.g., multivariate curve resolution, de Juan and Tauler, 2006), these spectra are more amenable for spectral matching.





Alternatively, taking the first $n$-th derivatives of the spectrum will remove the first $n$ terms of the $N$-th order polynomial and transform the rest of the signal (DeNoyer and Dodd, 2006). Since Gaussian (and most absorption) bands are not well-approximated by low order polynomials, they are not eliminated, i.e., their relative amplitudes and half-widths (ideally) remain unaffected by the transformation. This ensures that their value is retained for multivariate FT-IR calibrations (Weakley et al.,

2016). Moreover, derivative-based methods can improve resolution of absorption bands after transformation (illustrated in Figure 7). Derivative transformations can affect the signal-to-noise (S/N) ratio, however; inflating the relative contribution of small perturbations. Therefore, smoothed derivative methods such as the three-parameter Savitzky-Golay filter (Savitzky and Golay, 1964) are favored in order to minimize this effect and, in practice, only first and second derivatives are generally used with vibrational spectra to maintain a reasonable S/N ratio (Rinnan, 2014). In complex aerosol spectra caution must exercised

when interpreting the bands resolved by smoothed derivative filters since the filter parameters (i.e., bandwidth, kernel) all influence the outcome of the transformation. A major disadvantage of derivative filtering, in addition to the reduced visual connection to the original spectrum, relates to the inadvertent removal of broad absorption bands (Griffiths, 2006). Tuning filter parameters by trial-and-error may limit this type of band suppression to some extent. As a rule of thumb, the broad O-H stretches of alcohols (3650–3200 $\mathrm{cm}^{-1}$), carboxylic acids (3400–2400 $\mathrm{cm}^{-1}$), and N-H stretches of amines (3500–3100 $\mathrm{cm}^{-1}$)

are likely to be sacrificed as a result of derivative filtering (Shurvell, 2006). A willingness to balance this type of information loss against the simplicity and rapidity afforded by derivative methods must be considered in practice.

Different approaches have been used for processing of spectra for TOR calibration, including two interpolation and one derivative approach. Spectral processing is useful for spectra of PM collected on PTFE filters due to the significant contribution of scattering from the PTFE (McClenny et al., 1985). Small differences in filter characteristics lead to high variation in its

contribution to each spectrum; a simple blank subtraction of similar blank filters or the same filter prior to PM loading is not adequate to obtain spectra amenable for calibration (Takahama et al., 2013). As the magnitude of this variability is typically greater than the analyte absorbances, baseline correction models trained on a set of blank filters typically do not perform adequately in isolating the non-negative absorption profile of a new spectrum. Accurate predictions made by PLS without explicit baseline correction suggest that the calibration model is able to incorporate its interferences effectively within its

feature space if trained on both ambient samples and blank samples together, though visually interpretable spectra for general use is not necessarily retrievable from this model. For this purpose, models based on interpolation from the sample spectrum itself has been preferred. Takahama et al. (2013) described semi-automated polynomial and linear fitting to remove PTFE residuals remaining from blank-subtracted spectra, which was based on prior work for manual baseline correction by Maria et al. (2003) and Gilardoni et al. (2007). This correction method had been used for spectral peak-fitting, cluster analysis, and

factor analysis (Russell et al., 2009; Takahama et al., 2011) previously, and was used for 2011 IMPROVE TOR OC and EC calibration shown in Table 2 (Dillner and Takahama, 2015a, b; Takahama et al., 2016). Kuzmiakova et al. (2016) introduced a smoothing spline method which produced similar baseline corrected spectra (both visually and with respect to clustering and calibration) in ambient samples to the polynomial method without need for PTFE blank subtraction. While the non-analyte regions of the spectra are implicitly assumed, the flexibility of the local splines combined with an iterative method for

readjusting the non-analyte region effectively reduced the number of tuning parameters from four (in the global polynomial




approach) to one. The spline baseline method was used for TOR EC prediction in 2013 CSN (Weakley et al., 2018a). Second derivative baseline correction method was applied to 2013 CSN TOR OC calibration (Weakley et al., 2016).

Overall, differences in calibration model performance in TOR prediction between spline corrected and raw spectra models were minor for the samples evaluated in 2011 IMPROVE (results were comparable to metrics in Table 2). However, wavenum-
bers remaining after uninformative ones were eliminated (Section 3.3.2) differed when using baseline corrected and raw spectra — even while the two maintained similar prediction performance. Weakley et al. (2016) and Weakley et al. (2018a) used the Savitzky-Golay method and spline correction method for TOR OC and EC, respectively, in the 2013 CSN network, but did not systematically investigate the isolated effect of baseline correction on predictions without additional processing. A formal comparison between the derivative method against raw and spline-corrected spectra have not been performed, but this is an
area warranting further investigation. Standardizing a protocol for spectra correction based on targeted analyte is a sensible strategy, as spectral derivatives are associated with enhancement in specific regions of the spectra. The selection of baseline correction method may also consider the areal density of the sample, since the S/N is reduced with derivative methods. However, the success of derivative methods demonstrated for TOR OC in CSN samples (with systematically lower areal loadings than IMPROVE samples) indicates that the reduction in S/N is not likely a limiting factor for quantification in this application.
The derivative method appears to have significant advantage in reducing the number of LVs as demonstrated for TOR OC (Table 2). The derivative-corrected spectra model for 2013 CSN resulted in only 4 components in contrast to the 35 selected by the raw spectra model. While wavenumber selection and a different model selection criterion was simultaneously applied to the derivative-corrected model, a large reason for the simplification is likely due to the baseline correction. For reference, reduced-wavenumber raw spectra models for 2011 IMPROVE TOR OC and EC still required 7–9 components (the full-wavenumber
model required 15–28, depending on spectral baseline correction) (Takahama et al., 2016). A parsimonious model is desirable in that it facilitates physical interpretation of individual LVs as further discussed in Section 3.4.

The effect of baseline correction on reducing the scattering is illustrated by revisiting the TOR-equivalent OC predictions for the 2013 IMPROVE data set. Reggente et al. (2016) found that the raw spectra 2011 IMPROVE calibration model performed poorly in extrapolation to two new sites in 2013, particularly Fresno, CA, (FRES) and Baengnyeong Island, S. Korea (BYIS).
When using baseline corrected spectra, the median bias and errors are reduced from $0.28\,\mu g\,m^{-3}$ and $0.43\,\mu g\,m^{-3}$ and to $0.19\,\mu g\,m^{-3}$ and $0.28\,\mu g\,m^{-3}$, and $R^2$ increases from 0.79 to 0.91 for samples from these sites (Figure for baseline corrected predictions shown in Section 4.1.1). As the filter type remained the same, this improvement in prediction accuracy is likely due to the removal of scattering contributions in $PM_{2.5}$ particles in the new set that differs from the calibration set. Spectral signatures of nitrate and dust suggested the presence of coarse particles different than those in the 2011 calibration (and test)
set samples. 4.1).

### 3.3.2  Wavenumber selection

Wavenumber or variable selection techniques aim to improve PLS calibrations by identifying and using only germane predictor variables (Balabin and Smirnov, 2011; Höskuldsson, 2001; Mehmood et al., 2012). Typically, such techniques remove variables deemed excessively redundant, enhance the precision of PLS calibration, reduce collinearity in the variables (and





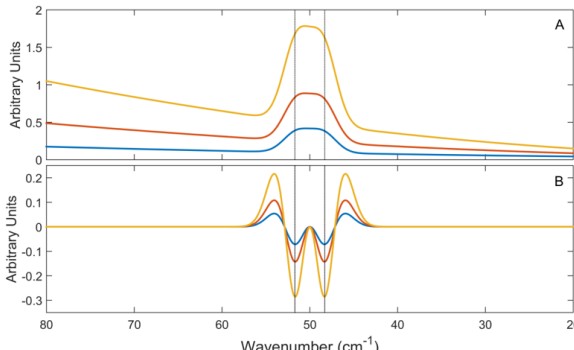

**Figure 7.** Three synthetic absorption spectra consisting of two unresolved Gaussian peaks superimposed on a polynomial baseline (A), and their 2nd order, 5-point, second derivative, Savitzsky-Golay filter transformations (B). Absorption spectra were constructed such that the additive, linear, and polynomial components ($a_N$) of the baseline scale with the amplitude of the absorption bands. The baseline is completely eliminated and the bands are better resolved as a result of filtering.

therefore model complexity) (Krämer and Sugiyama, 2011), and possibly improve interpretability of the regression. The simplest variable selection method based on physical insight rather than algorithmic reduction is truncation, in which regions for which absorbances are not expected or expected to be uninformative are removed a priori. Algorithmic variable selection techniques fall into three categories: filter, wrapper, and embedded methods (Saeys et al., 2007; Mehmood et al., 2012).

Filter methods provide a one-time (single-pass) measure of a variable importance with important and redundant variables distinguished according to a reliability threshold. Variables above such a threshold are retained and used for PLS calibration. Often, thresholds are either arbitrary or heuristically determined (Chong and Jun, 2005; Gosselin et al., 2010). In general, filter methods are limited by their need to choose an appropriate threshold prior to calibration, potentially leading to a suboptimal subset of variables.

The essential principle of wrapper methods is to apply variable filters successively or iteratively to sample data until only a desirable subset of quintessential variables remain for PLS modeling (Leardi, 2000; Leardi and Nørgaard, 2004; Weakley et al., 2014). Wrappers operate under the implicate assumption that single-pass filters are inadequate, requiring a guided approach to comprehensively search for the optimal subset of modeling variables. Since searching all $2^p - 1$ combinations of wavenumbers is not tractable for multivariate FT-IR calibration problems ($p > 10^3$), model inputs (or importance weights) are generally

randomized at each pass of the algorithm to develop importance criteria, foregoing an exhaustive variable search. Genetic algorithms and backward Monte Carlo unimportant variable elimination (BMCUVE) are examples of two randomized wrapper methods (Leardi, 2000; Leardi and Nørgaard, 2004). Wrapper methods generally perform better than simple filter methods and have an additional benefit of considering both variables and PLS components simultaneously during optimization. The major drawback to wrapper methods are generally longer runtimes (which may be on the order of hours for large-scale problems)

than filter methods.





As their name implies, embedded methods nest variable selection directly into the main body of the regression algorithm. For example, sparse PLS methods (SPLS) eliminate variables from the PLS loading weights ($w$), which reduce the number of non-zero regression coefficients ($b$) when reconstructed through eq. 5 (Filzmoser et al., 2012). The zero-valued coefficients obtained for each LV can possibly confer component-specific interpretation of important wavenumbers, but leads to a set of

regression coefficients which are overall not as sparse as methods imposing sparsity directly on the regression coefficients (Takahama et al., 2016).

Many methods select informative variables individually, but for spectroscopic applications it is often desirable to select a group of variables associated with the same absorption band. Elastic net (EN) regularization (Friedman et al., 2010) adds an L2 penalty to the regression coefficient vector in addition to the L1 penalty imposed by the least absolute shrinkage and selection

operator (LASSO) (Tibshirani, 1996), thereby imparting a grouping effect in selection. Interval variable selection methods (Wang et al., 2017) draw upon methods discussed previously but employ additional constraints or windowing methods to target selection of contiguous variables (i.e., an algorithmic approach to truncation).

Takahama et al. (2016) evaluated two embedded (sparse PLS) algorithms and one hyphenated method in which EN used as a filtering method prior to PLS calibration (EN-PLS, Fu et al., 2011) for TOR OC and EC calibration in the IMPROVE network.

A suite of reduced-wavenumber models were considered by varying model parameters that controlled the sparsity, and evaluated using cross-validation and separate test set samples. Since full-wavenumber calibration models (both raw and baseline corrected) for TOR OC and EC in the IMPROVE networks already performed well (Section 3.2.1), wavenumber selection did not improve model predictions but served mostly to aid interpretation of the most important absorption bands. Takahama et al. (2016) found that these methods could use as little as 4–9% of the original wavenumbers (2784 for raw and 1563 for spline

corrected) to predict TOR-equivalent OC and EC. EN-PLS consistently achieved the sparsest solution (by more than a factor of two in almost all cases) on account of the LASSO penalty applied directly to the regression vector. While all variable selection methods generally performed well for TOR-equivalent OC and EC prediction in 2011 IMPROVE samples, calibrations for organic functional groups built using sparse PLS algorithms appeared to be less robust in extrapolation to ambient sample spectra. While also being the most sparse, EN-PLS yielded similar predictions to the original PLS (full wavenumber) models

(Takahama and Dillner, 2015) that led to OC reconstruction from summed functional group contributions having better agreement with TOR OC than other sparse calibration algorithms, including EN without PLS. This finding suggests that variables eliminated for being uninformative in the calibration set samples may lead to undesirable oversimplification of a model that may be used with samples with potentially different composition, though this hypothesis has yet to be tested with calibrations developed with ambient measurements as reference, where the extent of extrapolation may not be so severe as with calibrations

developed with laboratory standards. Weakley et al. (2016, 2018a) applied BMCUVE to second derivative or spline corrected spectra in the CSN network. Improved MDL but otherwise similar performance metrics to the raw (full wavenumber) calibration model was obtained using the reduced model for TOR OC (performance described in Section 3.2.1), though the individual contributions of baseline correction and wavenumber selection to improvement in MDL was not investigated. The impact of wavenumber selection on model performance was not investigated for TOR EC, but the reduced-wavenumber model predicted

EC within TOR precision (Section 3.2.1). Interpretation of the selected wavenumbers are discussed in Section 3.4.



### 3.4 Interpretation

Interpreting the relationships among variables being used by a statistical model to make predictions is a challenging topic on account of their semi-empirical basis. In particular, it is possible to exploit statistical correlations among the variables to make predictions, which can be detrimental if the correlation changes or model is applied in a different context (further discussion is provided in Section 4.1.2). Therefore, model interpretation is strongly related to anticipation of model applicability. Inspection of how LVs and absorption bands are used by a model can give an indication of their importance, and possibly establish a physical basis between analyte concentrations and their relevant vibrational modes. Existence of sample subgroups and potentially influential subgroups can initiate indentification of relevant sample characteristics that have a disproportionate role in prediction. To some extent, discussions in Sections 3.1 and 3.3.2 focusing on eliminating uninformative variables (LVs or wavenumbers) during the model selection process is also relevant in this context (some of the same techniques are applicable to both tasks), but the focus will be on understanding the importance of the remaining variables. The importance of samples and specific attributes (concentration or composition) associated with them are addressed in Section 3.5.

As with complex mechanistic models, a general investigation can be carried out through sensitivity analyses (Harrington et al., 2000; Chen and Yang, 2011). One of the advantages of a PLS regression approach is that the contribution of each LV to response ($\boldsymbol{y}$) or spectra matrix ($\boldsymbol{X}$) can be characterized by the explained Sum-of-Squares ($SS$) and its normalized surrogate, Explained Variation ($EV$) (Martens and Næs, 1991; Abdi, 2010). The emphasis placed by a model on particular wavenumbers can be examined through its regression coefficients $\boldsymbol{b}$, Selectivity Ratio ($SR$) (Kvalheim, 2010), or the Variable Importance in Projection ($VIP$) metric (Wold, 1993; Chong and Jun, 2005). These quantities can be written using $j$ and $k$ as indices for wavenumber and LV (with $J$ as the total number of wavenumbers), respectively:

$$SS_{y,k} = q_k^2 \boldsymbol{t}_k^T \boldsymbol{t}_k \tag{8}$$

$$SS_{X,k} = (\boldsymbol{p}_k^T \boldsymbol{p}_k) \cdot (\boldsymbol{t}_k^T \boldsymbol{t}_k) \tag{9}$$

$$SS_{X,j} = \boldsymbol{p}_j \left(\boldsymbol{T}^T \boldsymbol{T}\right) \boldsymbol{p}_j^T \tag{10}$$

$$EV_{y,k} = SS_{y,k}/(\boldsymbol{y}^T \boldsymbol{y}) \times 100\% \tag{11}$$

$$EV_{X,k} = SS_{X,k}/(\boldsymbol{X}^T \boldsymbol{X}) \times 100\% \tag{12}$$

$$SR_j = SS_{X,j}/\left(\boldsymbol{e}_{X,j}^T \boldsymbol{e}_{X,j}\right) \tag{13}$$

$$VIP_{jk} = \left(J \frac{\sum_{\ell=1}^{k} SS_{y,\ell} \left(w_{\ell j}/\|\boldsymbol{w}_\ell\|\right)^2}{\sum_{\ell=1}^{k} SS_{y,\ell}}\right)^{1/2}. \tag{14}$$

These expressions apply to both calibration and new samples, though typically used for the former. Note that for new samples, the loadings ($\boldsymbol{q}$ and $\boldsymbol{p}$), sum-of-squares ($\boldsymbol{T}^T\boldsymbol{T}$) and the means used for centering of each array ($\boldsymbol{y}$ and $\boldsymbol{X}$) are fixed according to the calibration set. For PLS, the $EV_X$ is not as commonly examined as for other factor analysis techniques as the primary objective is in explaining the variation in $\boldsymbol{y}$. In addition to metrics characterizing the overall importance of latent and physical variables, the (normalized Euclidean) distance of individual samples from the center of the calibration space can be indicated by its leverage $h$. For mean-centered PLS, $h$ is computed for row vector of new scores $\boldsymbol{t}$ corresponding to sample $i$ weighted




by the inverse sum-of-squares of the calibration set (Martens and Næs, 1991):

$$h_i = \boldsymbol{t}_i \left( \boldsymbol{T}^T \boldsymbol{T} \right)^{-1} \boldsymbol{t}_i^T \qquad (15)$$

The sample leverage is used to assess influential points in the model, identify outliers, and estimate prediction variance (prediction intervals). Further discussion of leverage used in the last two objectives is discussed in Section 4.1. Regression coefficients
can oscillate between positive and negative numbers as higher number of LVs are used (Gowen et al., 2011) and their magnitude must be considered together with that of the absorbance (i.e., large regression coefficients coupled with small absorbances may not have a large impact on the modeled outcome), metrics such as SR or VIP can be more useful to assess their relative importance (the two vary in ease of interpretability for different types of data and data sets, Farrés et al., 2015).

For TOR analysis, VIP scores have been used to interpret wavenumber importance (Dillner and Takahama, 2015a, b; Weak-
ley et al., 2016, 2018a). VIP scores can also be used as a filtering method (Section 3.3.2) for wavenumber selection (e.g., Gosselin et al., 2010; Lin et al., 2013; Liu, 2014), but here they have been used only for post hoc interpretation for this work. The main principle is that the mean VIP score across all wavenumbers is unity, so those with more influence in explaining $\boldsymbol{y}$ carry values above and those with less influence fall below. However, Chong and Jun (2005) found that the actual importance threshold can be data-specific, with dependence on the proportion of uninformative predictors, predictor correlation, and the
actual values of the regression coefficients. Meaningful threshold values varied between 0.8 and 1.2 in the work of Chong and Jun (2005). VIP scores for TOR models are summarized in Figure 8. Wavenumbers associated with TOR OC not surprisingly spans a range of functional group structures. Common functional groups interpreted for both 2011 IMPROVE and 2013 CSN include aliphatic C-H and carbonyls (carboxyl, ketone, ester, aldehyde), with possible contributions from various nitrogenated (amine, amide, nitro) groups (Takahama et al., 2016; Weakley et al., 2016). Other candidate bonds are described but assigned
with less certainly on account of strong overlap of absorption bands in some spectral regions. Takahama et al. (2016) based their interpretation on the selected wavenumbers and VIP scores for both raw and baseline corrected models under a "common bond" that the two models are basing their prediction using the same set of functional groups rather than different ones. Based on this assumption, it appeared that the two models were using different vibrational modes (stretching or bending) for aliphatic C-H and alcohol O-H, though bending modes typically exhibit weaker absorption signatures. The capability to ac-
curately predict TOR-equivalent OC concentrations in samples with different OM/OC ratios (determined by functional group calibration models with FT-IR) as discovered through permutation analysis (Section 3.5.1) suggests that on average, there is some insensitivity to weighting of functional groups that determine the degree of functionalization in the sample.

For TOR EC, among other functional groups, wavenumbers selected between 1600–1500 $\mathrm{cm}^{-1}$ were attributed to C-C and C=C stretching in skeletal ring structures of aromatic or graphitic carbon (Takahama et al., 2016; Weakley et al., 2018a). While
this absorption band corresponds to lattice vibrations in graphitic carbon (Tuinstra and Koenig, 1970) and commonly used in Raman spectroscopy for characterization of soot particles (Sadezky et al., 2005; Doughty and Hill, 2017), a peak has been observed in mid-IR spectra only after crystalline structure is broken down through mechanical stress (Friedel and Carlson, 1971, 1972; Ṭucureanu et al., 2016). Nonetheless, a peak of moderate to broad width in this region is observed in soot (Akhter et al., 1985; Kirchner et al., 2000; Cain et al., 2010), soil black carbon (Bornemann et al., 2008; Cheng et al., 2008), and





coal (Painter et al., 1982). In constructing a PLS model to predict BC in soil by mid-IR spectra and PLS, Bornemann et al. (2008) further removed the potential influence of correlation between EC and OC in soil samples by predicting the BC content normalized by OC with an $R^2$ of 0.81. This analysis encouraged their interpretation that the aromatic structures visible in their first PLS loading weight vector were specific to BC, which potentially supports the same interpretation for atmospheric

samples. However, Weakley et al. (2018a) found that a calibration model for Elizabeth, NJ, did not require aromatic structures for prediction of TOR-equivalent EC. This site was characterized by high diesel PM loading, low OC/EC ratio, and low degree of charring compared to samples from other CSN sites in the 2013 data set. The calibration model was able to predict TOR-equivalent EC concentrations primarily using absorption bands associated with aliphatic C-H (also selected in the calibration model for the other 2013 CSN sites) and nitrogenated groups believed to be markers for diesel PM. A standard method for

quantification of soot (ASTM D7844-12, 2017) recommends the use of scattering characterized at $2000 \, \mathrm{cm}^{-1}$ (without baseline correction) on the assumption that there is no absorption usable for quantification. Given that baseline corrected spectra (in which scattering at $2200\text{-}1900 \, \mathrm{cm}^{-1}$ in addition to other wavenumbers with negligible absorption are forced to zero) are able to predict TOR-equivalent EC concentrations in both 2011 IMPROVE and 2013 CSN — and most relevant wavenumbers are in regions associated with visible absorption peaks — the predictions do not appear to be based on scattering in this application.

Early work by Pollard et al. (1990) reported a calibration for collocated EGA EC using a peak located at $666–650 \, \mathrm{cm}^{-1}$ in mid-IR spectra of PM collected onto PTFE filters at Glendora, CA. However, what vibrational mode this peak corresponds to is unclear, as there is also IR interference from the PTFE substrate in this region (Quarti et al., 2013). The true nature of operationally-defined TOR EC and a definitive reason that its concentration can be predicted from mid-IR spectra is an ongoing topic of investigation. Surface functionalization of graphitic combustion particle surfaces (Cain et al., 2010; Popovicheva et al.,

2014) are estimated to be a small fraction of the functional groups from organic aerosol in the same sample, and therefore considered to be unlikely to be useful for calibration. Soot emissions comprise both light-absorbing black carbon and organic carbon (Novakov, 1984; Petzold et al., 2013), and it is possible that both fractions exhibit mid-IR activity (some structures co-absorbing in the same region) that can be used for quantification. Whether the functional groups used for prediction of TOR-equivalent EC are due to the organic fraction associated with incomplete combustion or other indirect markers warrants

further investigation in controlled studies.

     While the large number of LVs used by the IMPROVE calibration models precluded attempt at identification of individual components, Weakley et al. (2016) was able to do this for 2013 CSN TOR OC calibration models on account of their low complexity. Application of second-derivative baseline correction, BMCUVE wavenumber selection, and model selection by Wold's $R$ criterion resulted in a 4-LV model for TOR OC. Further nuanced interpretation was aided by re-projection of LVs

onto PCA space which modeled much of the same variance as PLS scores, but were formulated and arranged according to their capability to explain the remaining variance in the spectra instead of the covariance with respect to TOR OC. By visualizing the sample spectra in two dimensions of this space using a conventional biplot, Weakley et al. (2016) identified a subset of samples with extraneous variance in 2013 CSN spectra attributed to water vapor in the beam path present during spectra acquisition in the laboratory. While the water vapor conferred minimal prediction error, loading this spectral interference onto

one dimension and excluding it the final calibration model improved interpretability with a more parsimonious model using





only the 3 remaining components. Surprisingly, a single component representing an organic mixture explained close to 90% of the TOR OC variance, with the remaining two components attributed to interferents: PTFE substrate and ammonium nitrate (explained variation of 3–4 % each).

Model interpretation is a continual challenge but a necessary aspect of statistical modeling from a chemometrics perspective, and remains an active area of investigation for TOR analysis. While the LVs are not constrained to be non-negative as factors for Multivariate Curve Resolution, Positive Matrix Factorization and Non-negative Matrix Factorization (Paatero, 1997; Lee et al., 1999), the relative variation of scores can be analyzed alongside auxiliary measurements to identify their importance toward specific PM samples. This association can be made in a correlative capacity (Russell et al., 2009; Faber et al., 2017), or through more sophisticated means such as target transformation factor analysis (Henry et al., 1984; Hopke, 1989). In addition, the way of obtaining LVs can be modified to accommodate features from TOR OC and EC simultaneously. A variant of PLS that can potentially aid in this endeavor is "PLS2", which uses a shared representation of LVs for multiple response variables (Martens and Næs, 1991). Shared representations are commonly used in multi-task learning (Caruana, 1997) to build models that generalize from fewer, diverse training instances, and may additionally confer benefit in this context for understanding the inter-relationship between these two substances and their thermal fractions. The univariate-response formulation of PLS ("PLS1") as described in Section 3.1 has been the focus of past work with TOR calibrations as it typically achieves the same or better accuracy as PLS2 with fewer LVs (Martens and Næs, 1991), but the potential for PLS2 in improved interpretation and robustness in a wider range of contexts is an area that can be further explored.

## 3.5 Sample selection

To design a campaign to collect both FT-IR spectra and reference measurements or to select among available collocated measurements in a database to construct a new calibration model, it is necessary to address the question of *how many of which type of samples do we need*? Provided that the form of a data set can be fit by several models, it is possible for the simpler ones with more training data to outperform more complex ones with less training data for new predictions (Halevy et al., 2009). This argument can be rationalized in a chemometric context by conceptualizing an ideal calibration model as one built upon samples of identical composition and concentration (with replicates) for every sample in the prediction set. Especially for complex PM components such as TOR OC and EC that have a multitude of absorption bands in the IR from both target and interfering substances, enough samples must be included in the calibration set to span the range of multiple attributes. For each unique sample removed from the calibration set, the corresponding composition in the prediction set must be estimated by mathematical interpolation or extrapolation from the remaining samples. Reducing the number of calibration samples increases the dependence of the predictions on the functional form or weighting scheme (with respect to variables and samples) of the selected model with possible consequences for prediction accuracy. Lacking mechanistic constraints, predictions from data-driven models may exceed physical limits with increasing reliance on the underlying algorithm over measurements. The obvious importance of chemical similarity in calibration can be related back to physical principles that give rise to the observed mid-IR spectrum. First, for any given wavenumber, the absorption scales with analyte abundance — simpler calibration models in analytical chemistry built on this principle dictate that the concentration range covered by





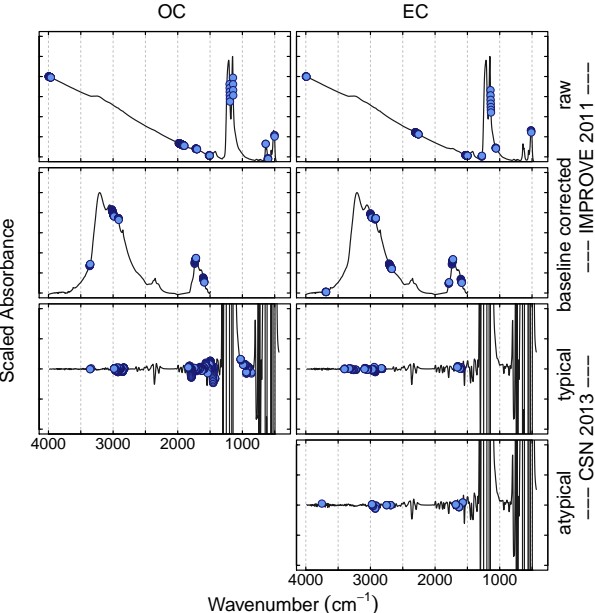

**Figure 8.** Selected wavenumbers (blue points) overlayed on mean of calibration spectra (black lines). 2011 IMPROVE spectra remain unprocessed ("raw") or baseline corrected using smoothing splines ("baseline corrected"), while the 2013 CSN spectra are baseline corrected using the Savitsky-Golay 2nd derivative approach. "Atypical" and "typical" categories for 2013 CSN EC refer to samples for Elizabeth, NJ, and the remaining nine sites, respectively.

calibration samples should bound the concentrations in the new samples so that values are interpolated rather than extrapolated to minimize prediction error. Second, complex absorption profiles arise from heterogeneous broadening of absorption bands in the condensed phase. Therefore, samples with a similar chemical composition to new samples are likely to have similar patterns of absorbance and interferences that can be accounted for by the calibration model.

5    A basic premise follows that calibration models built with samples having similar spectroscopic profiles, specifically near the most relevant absorption bands, are likely to yield better prediction results for new smaples. For analysis of simple mixtures, one common strategy pursued in experimental design is to prepare samples that populate the chemical coordinates (e.g., molar concentrations of its constituent species) of anticipated composition according to Euclidean distance (Kennard and Stone, 1969). However, this procedure does not guarantee that the training and prediction data will have similar distributions in the

10   feature space of an effective calibration model (i.e., similarity may not be best characterized by Euclidean distances). This task is further complicated by the fact that chemical similarity is not easy to define for composite substances (TOR OC) or chemically ambiguous quantities (TOR EC). Moreover, the samples for calibration at the level of chemical complexity of atmospheric mixtures are typically limited by the availability of collocated measurements (e.g., TOR reference measurements together with sample spectra from PTFE filters).





In the context of these challenges, the canonical ("base case") strategy for TOR OC and EC calibration has been to use space and time as a proxy for composition. A stratified selection approach — selected samples are evenly spaced out over a full year at each measurement site — is used construct the calibration set, as there is reasonable expectation that an adequate representation of emission sources and extent of atmospheric processing can be captured. Blank PTFE filter spectra are added

to the calibration set and their corresponding reference concentrations are set to zero, as this value is equally valid to the TOR-determined concentration for below-MDL samples. Excluding irregular events (e.g., wildfires), this approach can be effective in building a general calibration model for atmospheric samples and has demonstrated performance. We further summarize our efforts in understanding which types of samples are important (Section 3.5.1) and how many samples are needed (Section 3.5.2). Lastly, we describe how specialized calibration models can better serve a specific set of samples that are not well-

represented in the feature space of all calibration samples (Section 3.5.3).

### 3.5.1   Important attributes

Our findings indicate that many, though not all, methods for sample selection can lead to an acceptable calibration model as determined by evaluation criteria described in Section 3.2. To investigate which aspects of similarity are important in this regard, Dillner and Takahama (2015a; 2015b) performed permutation analyses on the available set of samples to study how

differences between calibration and test set samples influenced prediction errors. Samples were grouped according to values of descriptors chosen to capture the effect of analyte concentration (TOR OC, EC), source and degree of functionalization (OC/EC and OM/OC), and inorganic interferences (ammonium/OC, ammonium/EC). Predictions were evaluated when the distribution of these descriptors represented in the calibration set were selected to be either similar or different to those in the test set. To construct calibration and test sets according to these specifications, samples were arranged in order of a particular

attribute. For similar calibration and test set distributions, every third was reserved for the test set while the remainder was used for calibration. To examine the effect of extrapolation with respect to any attribute, the calibration set was constructed from samples with the lowest two-thirds or highest two-thirds of attribute values, and the remainder used for the test set. To examine the effect of interpolation, the highest third and lowest third were used for calibration and predictions made on the middle third of samples. Inadequate representation of any of these variables in the calibration set led to increased errors in

model predictions, but with typically low bias in interpolation. TOR OC could be predicted with only marginal increase in bias (median absolute bias of 0.1 $\mu g\,m^{-3}$) and no increase in normalized error ($\sim$10%) even when extrapolating predictions on average three times higher, indicating a calibration that was effectively linear over the range tested (0–8 $\mu g\,m^{-3}$). For samples varying in OM/OC ratio between 1.4–2.5, normalized error in predicted TOR OC increased from $\sim$10% when the calibration and test sets were similar to 14–17% when they were forced to diverge according to the segmentation described above, but

the predictions remained unbiased. The largest increase in prediction error came when using calibration samples with low ammonium interference (low ammonium/OC ratio) to high ammonium content, with an increase in normalized error of from $\sim$10% to 24%. For TOR EC, almost every extrapolation scenario resulted in an increase in either bias or normalized error (by 10 to 60 percentage points), suggesting its sensitivity to a large number of sample attributes.





Such permutation analyses permit independent evaluation of attribute importance only to the extent that they are not corre­lated in the samples. For instance, for 2011 IMPROVE, much of the variability across the entire data set was driven by the two collocated urban sites in Phoenix, AZ, which contained higher concentrations of less functionalized PM in general than the remaining rural sites. However, normalization strategies — e.g., of ammonium by OC or EC — reduced confounding effects.

Dillner and Takahama (2015a; 2015b) only tested each univariate case in turn, but multidimensional permutation analysis in which samples are partitioned according to differences across multiple variables for model building and testing may be possi­ble with a large number of samples. Computational resources permitting, bootstrap sampling combined with post analysis may provide another means of testing the importance of particular attributes in such instances.

### 3.5.2 Number of samples

The minimum number of samples required by a model is dependent on the capacity of its calibration samples to collectively represent the diversity of composition in new samples, and the algorithm to effectively interpolate or extrapolate into unpopu­lated regions of the composition space. To illustrate this notion, we present the change in prediction metrics for TOR-equivalent OC as a function of the number of ambient samples in the calibration set (Figure 9). Beginning with samples selected accord­ing to the base case strategy (stratifying by space and time) as the initial reference, the number of ambient samples in the

calibration set are reduced while the number of blank samples are held constant. The set of test samples are also fixed for all evaluations. While the conclusions are not strikingly obvious, some overall trends can be noted. Figure 9 shows a general decrease in prediction accuracy with fewer number of ambient samples, especially below ∼150 samples, though individual differences among most models are not statistically significant. The gradual degradation in prediction accuracy is attributed to difficulty in maintaining representativeness of important attributes with a small number of samples. Figure 10 shows the

increasing difference in empirical probability distributions of attributes in the calibration and test set samples as a function of the number of ambient samples using the Kolmogorov-Smirnov test statistic (higher values indicate higher dissimilarity between the calibration and test set distributions). The increase in differences between the distributions in TOR OC, but par­ticularly the ammonium/OC ratio, is the primary cause as it was determined to be a critical attribute for TOR OC prediction (Section 3.5.1). Due to the diminising statistical power with fewer calibration samples, statistical significance is not established

in this regime; we therefore interpret these results qualitatively. The MDL is generally maintained or improved with decreasing number of ambient samples, which is sensible as the number of blank samples grows in proportion. On the other hand, the number of blank samples (varied between 0 and 36) when included with 501 ambient samples in the calibration set (Dillner and Takahama, 2015a, b) did not have a large effect on the MDL.

We might conclude that larger calibration sets that more likely cover the range of attributes in new samples might lead

to better model performance. Reggente et al. (2016) shows an example for raw spectra. Without baseline correction, TOR OC concentrations for two sites — FRES and BYIS —in 2013 IMPROVE were not predicted well by the original model. Predictions were shown to improve when samples from these sites were included (Reggente et al., 2016). In this case, the calibration set without Fresno and Korea was too small in that it did not contain the appropriate representation of specific sample characteristics.However, as with wavenumbers, populating the calibration set with increasing number of unrelated



or uninformative samples with respect to a targeted class of samples may lead to added noise or bias from unfavorable model weighting. In such instances, smaller, dedicated models may be better for specific classes of samples provided that it is possible to distinguish which model is best suited for each sample. In the next section, we describe cases in which a smaller subset of samples for calibration have been found to be appropriate for improving specific performance targets.

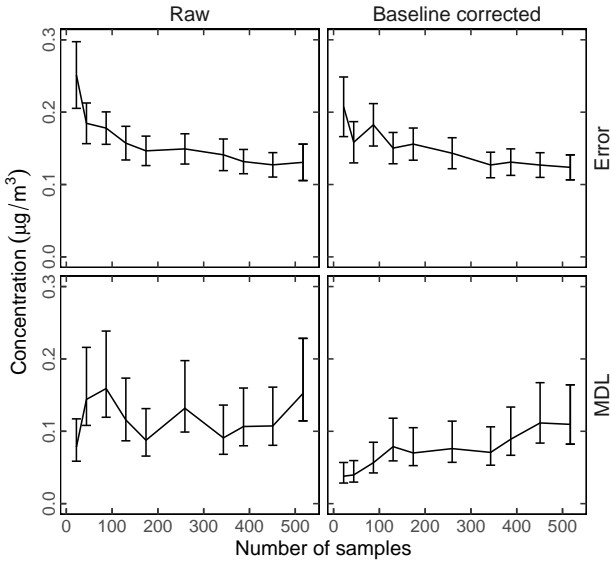

**Figure 9.** The prediction accuracy for TOR OC as a function of the number of ambient samples in the calibration set (the number of blanks were kept constant at 36). Using the 2011 IMPROVE base case calibration model, every $n$th sample was removed (which leaves spatial and temporal representation of samples close to the original set). The performance metrics are computed on the same 286 test set samples for all calibration models.

### 3.5.3 Smaller, specialized models

While a large, monolithic model may be most capable of accommodating diverse composition in prediction set samples, models that assume underlying structure of the chemical domain for interpolation or extrapolation may be susceptible to undue influence by one or more groups of (high leverage) samples and return biased predictions for a specific set of underrepresented samples. Statistical localization is the process by which calibration models are built with samples that are closest in composition to samples for which predictions are desired. While the overall number of samples used for training in each localized model is reduced, the distribution of the calibration model better reflects that of the subset of samples for which new predictions are to be made. Together with a classifier capable of selecting the appropriate localized model for each new spectrum, several models can collectively function as a single multilevel model to provide a best estimate of the targeted concentration.

This approach has been applied to TOR EC calibration in both networks studied (Dillner and Takahama, 2015b; Weakley et al., 2018a) (Figure 11). Dillner and Takahama (2015b) constructed a multilevel model consisting of calibrations for two





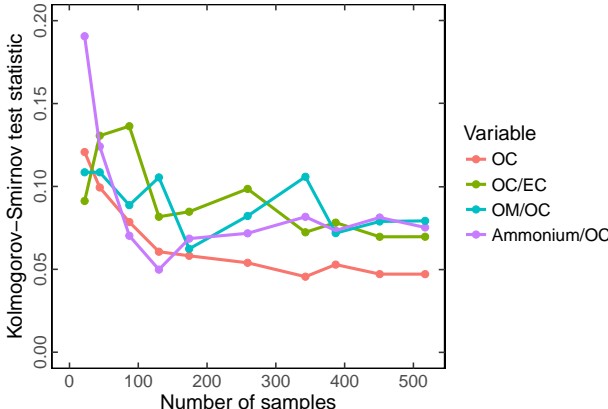

**Figure 10.** Kolmogorov-Smirnov (KS) test statistic for different number of calibration samples used in Figure 9. The KS statistic characterizes the difference between two empirical distribution functions; in this case determined for probability distributions of each variable between the calibration and test set samples.

different concentration regimes for 2011 IMPROVE. A calibration model using only a third of the lowest concentration samples (areal density $<0.68\,\mu\mathrm{g\,cm^{-2}}$) led to an MDL of 0.01–0.02 $\mu\mathrm{g\,m^{-3}}$, while using the full range of areal loadings for calibration led to an MDL of 0.03–0.08 $\mu\mathrm{g\,m^{-3}}$. Overall prediction errors for low samples were also reduced with a dedicated model, but to a lesser extent than the MDL. The full range model served as a classifier; predictions that fell below the areal loading

threshold according to this model were refined with the low-concentration calibration model. As discussed in Section 3.4, Elizabeth, NJ, (ELLA) was believed to be influenced by diesel emission sources that led to different PM composition and spectral characteristics from the remaining nine CSN sites. Therefore, predicted concentrations for ELLA were systematically biased low compared to observations. Weakley et al. (2018a) trained a partial least squares discriminant analysis (PLS-DA) model on geographical location to segregate typical samples from atypical ones that resembled ELLA spectra. Spectra classified

as being atypical were predicted using a model trained solely on ELLA samples, while the ones classified as typical were predicted using a model trained on the rest of the samples. Considering the overall model performance for all samples, using this multilevel approach led to an improvement in $R^2$ from 0.76 to 0.88, and a decrease in bias from 5.2 to 2.7% (with corresponding improvements in MDL, precision, and other figures of merit). The difference in metrics were largely due to improvement in ELLA predictions, as the predictions for non-ELLA samples were similar in both approaches (mean errors of

0.15 and 0.16 $\mu\mathrm{g\,m^{-3}}$, and $R^2$ of 0.83 and 0.85 for the monolithic and multilevel model, respectively).

## 4    Operational phase of a calibration model

The operational phase of the model marks a departure from the building and evaluation phase (Figure 2) in that reference measurements may no longer be available on a regular basis. Within this constraint, we must continually monitor the performance



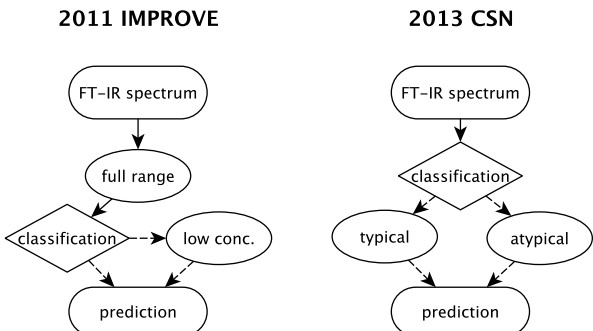

**Figure 11.** Multilevel modeling strategies used for TOR EC in the IMPROVE and CSN network. In the left figure, "full range" denotes the calibration model using the full range of TOR EC concentrations, while the "low conc." denotes the model using only the lowest third. In the right figure, "atypical" samples were taken from a particular site (ELLA) while the "typical" samples comprised the rest.

of the model by introspective means, and update the calibration as necessary. To this end, we describe methods for anticipating prediction errors arising from precision and bias (Section 4.1), and strategies for calibration maintenance (Section 4.2).

## 4.1 Anticipating prediction errors for new samples

We dedicate this section to describe ways for anticipating prediction errors in new samples during the operational phase of a
calibration model. Higher prediction errors may arise from a decrease in precision, or additional biases incurred for samples that are not well-represented by the calibration samples. The former can be approximated from the measurement noise characterized from the calibration set, while the latter is assessed on a more qualitative scale based on similarity of new samples to those in the calibration set. Anticipating these errors is imperative for reporting estimated precision for new samples, monitoring systematic changes in model performance, and selecting an alternate calibration model for new samples when prediction
quality is questionable. For this task, we assume the unavailability of reference measurements for which evaluation methods in Sections 3.2.1 and 3.2.2 would otherwise apply; and primarily rely on spectral characteristics. To this end, Section 4.1.1 discusses the construction of prediction intervals around point estimates, Section 4.1.2 covers the strategy for outlier detection, and Section 4.1.3 illustrates the use of sample similarity assessment for comparing suitability of models. The raw-spectra TOR EC calibration model for IMPROVE 2011 introduced by Dillner and Takahama (2015b) and evaluated for 2013 by Reggente
et al. (2016) is revisited on account of its high prediction error and difficulty anticipating prediction errors compared to TOR OC.

### 4.1.1 Sample-specific prediction intervals

In Section 3, discussions focused around providing and evaluating point estimates of prediction. Additionally, interval estimates for each sample can be obtained to determine prediction uncertainty under a fixed relationship between model and data assumed





under conditions of the calibration. In effect, prediction intervals describe magnitude of errors that are similar to those in the calibration set, and can be obtained from error propagation or resampling (bootstrap or jacknife) (Olivieri et al., 2006), or by employing a Bayesian framework (Murphy, 2012). We will restrict our discussion to estimating prediction intervals as they pertain to multivariate linear regression (including PLS). Provided that sufficient data exists, numerically-resampled intervals

can be generated free of assumptions regarding underlying distributions, but the error propagation approach is favored on account of its connection to the fundamental processes contributing to the errors. The standard error of prediction has two primary contributions: the model contribution from calibration, and the measurement contribution from the prediction sample. These contribute nonlinearly to the prediction error, but an approximate expression can be derived through local linearization (i.e., neglecting higher-order terms typically assumed in error propagation) (Phatak et al., 1993; Denham, 1997; Faber et al.,

2003; Serneels et al., 2004). This approximation results in a tractable expression for the rediction standard error $\sigma_{\hat{y},i}$ similar to that used by ordinary least squares regression, but considers heteroscedastic errors (Faber and Bro, 2002; ASTM E1655-17, 2017):

$$\sigma_{\hat{y},i} = s(1 + h_i)^{1/2} \ . \tag{16}$$

The point estimate of prediction can then be bounded by an interval defined as $\pm t_{\alpha,\nu}\sigma_{\hat{y},i}$, where $t_{\alpha,\nu}$ denotes a $t$-distribution

with significance level $\alpha$ and degrees of freedom $\nu$. $s$ is estimated from the fitting error — the mean squared error of calibration (MSEC: squared error normalized by the degrees of freedom). While a common assumption is that $s$ captures only the prediction variance, the MSEC can implicitly include the prediction bias if present in the fit of the calibration set. $h$ is the leverage introduced in eq. 15, and its role can be rationalized by the fact that samples closer to the "average" calibration sample are more precisely estimated than those which are further away. The approximations made for eq. 16 results in a method that is

most applicable for small noise and small range of FT-IR absorbances (Faber and Kowalski, 1997a, b). Furthermore, prediction standard error can be refined by subtracting the precision of the reference measurement (Faber and Bro, 2002; Faber et al., 2003), but is not considered here.

The prediction intervals given by eq. 16 calculated for TOR-equivalent OC and EC are shown in Figure 12. Low standard errors of predictions anticipate low prediction errors, but prediction errors for higher concentrations (3–85 µg cm$^{-2}$) are more

variable than indicated by the precision error. While deviations from observations in calibration are mostly explained by eq. 16, Reggente et al. (2016) and Weakley et al. (2018a) found that actual prediction errors do not always scale with computed leverage. This phenomenon is also reported in other applications (Zhang and Garcia-Munoz, 2009), and indicates the possible role of bias due to differences in composition that is not well-captured by this metric.

It is also relevant to consider the standard errors of prediction for the TOR measurements (Chow et al., 2007). Naïve propaga-

tion of reported errors across the relevant thermal fractions (including pyrolized carbon) leads to estimates of relative precision that approach 7 and 14% for TOR OC and EC, respectively, for the highest concentrations observed for this IMPROVE data set. As the errors are not truly independent for each sample, a simple summation of prediction variances may lead to an overestimation. However, these calculated errors are close in magnitude to the average collocated precision error estimated for 2011 IMPROVE (15 and 23% for TOR OC and EC, respectively, Table 2), and the combined uncertainty estimated from analytical,





cross-laboratory, and cross-sampler effects (Brown et al., 2017). The relative precision estimated for their respective calibration models using eq. 16 converges toward values which are approximately 3 times lower for both variables. The standard errors of prediction of a multivariate model can be lower than the reference measurements from which it is derived, as random errors from the latter are averaged out in the calibration process — especially when a large number of calibration samples are used

5 (Difoggio, 1995). However, given that the apparent collocated precision for model predictions are on a par with TOR (Table 2), it is likely that uncertainties calculated from eq. 16 are underestimated on account of unaccounted variations. Nonetheless, a general conclusion can still be drawn that many samples are predicted within uncertainty. There remain a samples (167 for TOR OC and 126 for TOR EC, out of 2177 total) that can be identified (in red, Figure 12) as having prediction errors which fall outside the anticipated range of uncertainty of both model and measurement. We describe procedures for algorithmically

10 detecting these samples in the absence of reference measurements in Section 4.1.2.

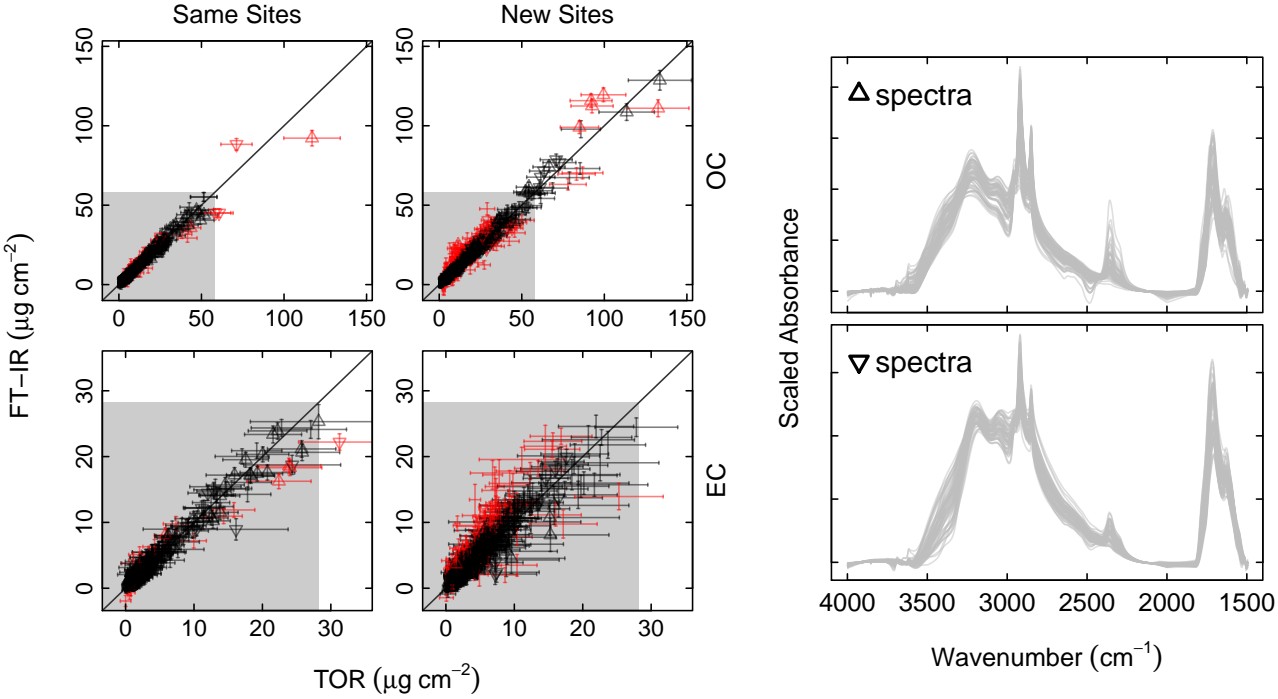

**Figure 12.** Point estimates and prediction intervals for the TOR-equivalent concentrations in the 2013 IMPROVE prediction set. Gray shades indicate extent of areal mass densities in the calibration samples. Triangles represent samples associated with wildfire burning (scaled spectra shown in right column). Red samples correspond to those for which the difference between predicted and observed concentrations exceed the combined uncertainties at the $\alpha = 0.05$ significance level.





### 4.1.2 Outlier detection

As described in Section 3.5, a calibration model that is likely to be suitable for a new sample is that which is trained on samples with similar concentration and composition. Therefore, identifying samples which are different from those in the calibration set of a particular model is closely tied to anticipation of potentially high prediction errors due to incurred bias. We first review

possible categorizations of samples in a Venn diagram (Figure 13). Within a multivariate space encompassing all samples, some will lie at the edge of the domain (*extreme values*), while others will lie in sparsely populated regions of the interior (*inliers*). Some of these extreme values and inliers will be statistically surprising given the rest of the points, and are typically labeled as *outliers* or *anomalous samples* (Barnett and Lewis, 1994; Jouan-Rimbaud et al., 1999; Aggarwal, 2013). We note that inliers are sometimes used to refer to statistically different samples which lie within the composition domain, but we reserve the word

outlier for all statistically significant samples in this paper. New samples in furthest proximity from calibration samples in this composition space require aggressive extrapolation or interpolation (i.e., they are least constrained by data), and are most likely to suffer in prediction performance. However, the actual increase in prediction error (if any) will depend on the functional relationship among variables and how well they are represented by the model — e.g., a linear relationship modeled by a linear mapping may perform adequately in interpolation and extrapolation. Therefore, not all outliers may be associated with high

prediction errors.

Dissimilarity can be expressed as a measure of distance or a discrete label of normal/anomalous resulting from an unary (one-class) classification (Brereton, 2011). Identification of dissimilar observations is the subject of many disciplines including chemometrics, machine learning, and statistical process control and are referred under various names: anomaly detection, fault detection, novelty detection, and outlier detection (e.g. Wise and Gallagher, 1996; Montgomery, 2013; Pimentel et al., 2014).

Together with knowledge regarding "prediction outliers" (samples with surprisingly high prediction errors), decisions can be grouped into the following outcomes (Figure 13): True Negative (TN; samples are classified as being similar and prediction error is low), True Positive (TP; samples are classified as being dissimilar and prediction error is high), False Negative (FN; samples are classified as being similar while prediction error is high), and False Positive (FP; samples are classified as being dissimilar while prediction error is low). The realization of these outcomes by a classifier can be used to judge its performance.

We note that in contrast to the multilevel modeling strategy described in Section 3.5.3, the problem of error anticipation is to build a classifier that identifies all samples not similar to those in the training set (i.e., outliers, some of which may have anomalously high magnitude of prediction error) without exhaustive knowledge or separate training sets comprising the new sample types.

Without reference measurements, many external indicators might be used to characterize differences between new samples

and those in the calibration set. For instance, the fraction of inorganic to total PM may give an indication of ammonium to OC ratio, or $NO_x$ may be a valid surrogate for EC in many urban situations. However, our primary objective is to rely on indicators of composition and concentration that can be extracted directly from the FT-IR spectrum to determine the appropriateness of an existing calibration model to the new samples. Baseline corrected spectra have been used in the past to characterize similarity among ambient aerosol spectra through cluster analysis (e.g., Takahama et al., 2011; Ruthenburg et al., 2014), and





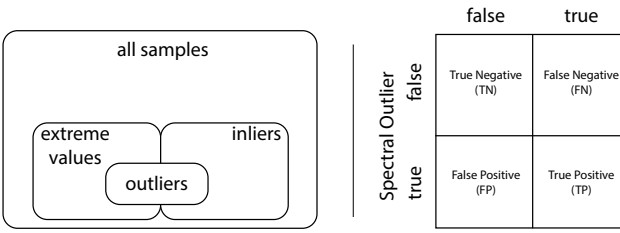

**Figure 13.** Venn diagram (not to scale), left, and confusion matrix, right, depicting the relationship between detected outliers and magnitude of prediction errors.

can also be used for classification (Fearn, 2006; Isaksson and Aastveit, 2006). For instance, many of samples with large deviations in predictions of TOR-equivalent OC from observed values are spectroscopically similar (Figure 12) and exhibit sharp methylene peaks and large carbonyl absorbances present in spectra of biomass burning samples (Hawkins and Russell, 2010; Russell et al., 2011). Locations and dates of some these samples are consistent with known periods of wildfires, and

will be the topic of future investigation. The underrepresentation of these types of samples in the 2011 IMPROVE calibration (and test) sets, or simply the higher concentrations beyond the calibration range may explain the proportionally high prediction errors incurred for these samples. The highest TOR EC concentrations in 2013 are associated with FRES, an urban site, and BYIS, an international site, both of which were not part of the 2011 calibration set. Spectral matching combined with model interpretation (Section 3.4) can identify particular sample types that may be problematic for a calibration model. However,

as sparse calibration modeling has shown (Section 3.3.2), not all spectral features are likely to be relevant for prediction of TOR OC or EC concentrations. Therefore, transformations specific for the target analyte (which can include but are not limited to spectral processing techniques described in Section 3.3) are likely to reveal the discriminating spectral features for distinguishing samples that are different from those in the calibration set.

Projection of the spectra in the feature space of the calibration model (i.e., factor scores and residuals of PLS or PCA, kernel

distances, latent encoding in Gaussian process) after appropriate spectra processing and wavenumber selection can provide spectral comparisons that are specifically meaningful for prediction of the response variable (Nomikos and MacGregor, 1995; MacGregor and Kourti, 1995; Camci et al., 2008; Ge and Song, 2010; Serradilla et al., 2011). For PLS regression, the feature vectors (scores) can be combined into a single metric called the Mahalanobis distance (Mahalanobis, 1936) or Hotelling's $T^2$ statistic (Hotelling, 1931), which are both proportional to the leverage introduced in eq. 15. The two terms are often used

synonymously (e.g., Kourti and MacGregor, 1995; ASTM E1655-17, 2017), but can also be defined differently according to rank approximation of $\boldsymbol{X}$ or a coefficient making the $T^2$ comparable to the $F$-distribution (e.g. De Maesschalck et al., 2000; Brereton and Lloyd, 2016; Brereton, 2016). We will adopt the convention of defining $T^2 \equiv D_M^2$, but reserve Hotelling's $T^2$ statistic for use with its eponymous test to determine out-of-limit samples (e.g., in statistical process control) and $D_M^2$ for a general distance measure (which is also used in classification methods built upon different criteria). Outside of this feature

space, the $Q^{(X)}$-statistic estimated using residuals $\boldsymbol{E}$ of spectra reconstructed from its latent variables (eq. 7) (Jackson, 2004)





can additionally indicate variations orthogonal to the feature space, and hence variations which are orthogonal to the modeled portion of the response variable (Höskuldsson, 1996; Bro and Eldén, 2009). Therefore, $Q^{(X)}$ is typically monitored over time alongside $T^2$. The two metrics for mean-centered PLS can be written as follows:

$$T_i^2 = D_{M,i}^2 = (N-1) \cdot h$$
$$Q_i^{(X)} = e_{X,i} e_{X,i}^T = x_i \left( I - P P^T \right) x_i^T$$

$N$ is the number of samples in the calibration and $h$ is the leverage from eq. 15. $P$ is the matrix of loadings (eq. 6) and $e_X$ denotes the row vector of residuals associated with each sample (eq. 5), equivalent to the product of latent variables unused for calibration. In an analytical chemistry context, high values of $T^2$ result from extreme values or unusual combinations of the same chemical components as those in the calibration set, whereas introduction of new analytes or interferences that result in
spectroscopic response lying outside of the modeled domain would be carried in the residuals (Wise and Roginski, 2015). In practice, the separation of unfamiliar contributions to the spectra is likely not as clean, particularly with respect to nonlinear phenomena (e.g., scattering) which can be spread over multiple factors, and the portion of the spectroscopic signal associated with new substances may not be entirely apportioned to the residuals.

For classification purposes, thresholds for $T^2$ and $Q^{(X)}$ are determined from the $F$ distribution and $\chi$-square distribution,
respectively, at different significance levels (Kourti and MacGregor, 1995). Classification and dissimilarity characterization by $T^2$ for a given data set performs best when the points converge toward a multivariate normal distribution. Such a distribution becomes less representative of the data set when the problem increases to proportions of extremely high dimensionality, where points become sparsely dispersed throughout the vast composition space rather than clustered around a single centroid (Domingos, 2012). To alleviate this problem, it is useful to conceptualize different relationships of training data in the column
space of $T$ and $E$ against which new samples are compared. This task can be fulfilled by unary (one-class) classifiers that learn patterns from the data without imposition of global structure (e.g., normality). These approaches may employ superposition of local potential or kernel density functions (Jouan-Rimbaud et al., 1999; Latecki et al., 2007), kernel methods (Schölkopf et al., 1999), or recursive partitioning of the chemical space (Liu et al., 2008) for detection of points separated from the from the remainder of the samples.

For the 2013 IMPROVE data set, Reggente et al. (2016) used the 2011 IMPROVE calibration models developed by Dillner and Takahama (2015a; 2015b) and applied the Mahalanobis distance metric. Heuristic thresholds for $D_M^2$ and the prediction error were determined as their respective maximum values in the 2011 IMPROVE test set for purposes of classification. The number of samples in 2013 which had prediction errors greater than the selected threshold was small for both TOR OC and EC — out of 2177 paired samples across 17 sites (and analytical blanks), only 36 (TOR OC) and 22 (TOR EC) samples (1–2%
of total) were determined as having high-errors according to this criterion. The overall accuracy (fraction of TN and TP out of total) was high, with 98% for both TOR OC and EC. These numbers are enviable for any classifier but was largely aided by the low number of high-error samples, which resulted in high overall accuracy from a permissive $D_M^2$ threshold and a limited number of FP classifications. When considering prediction intervals of both prediction and reference measurement, some of these high prediction errors are within anticipated uncertainties of the samples, while a few anomalous samples with errors





outside of the range of uncertainties occur with lower absolute prediction errors (Section 4.1.1 and Figure 12). Therefore, we first correlate the results of outlier analysis to samples with prediction errors that lie outside of expected agreement (i.e., prediction outliers). We then revisit the topic of using these classification algorithms to identify samples with the highest magnitude of prediction errors.

For this discussion, it is useful to define two additional metrics: True Positive Rate (TPR) is the fraction of samples with high error correctly identified as such, and the False Positive Rate (FPR) is the fraction of samples with low errors that are incorrectly identified as having high error. In a coordinate space with TPR as the ordinate and FPR as the abscissa (Figure 14), the perfect model lies at (0, 1). For detecting new or anomalous spectra, we explore classifiers introduced above (potential function method, one-class SVM, and isolation forest) and consider their tradeoffs in TPR, FPR, and overall accuracy. For the

potential function method, the radial basis function (RBF) is selected; the free parameters are the number of nearest neighbors used to determine the kernel width parameter and the confidence level for the thresholds. For one-class SVM, the RBF kernel is also used with the kernel coefficient and effective thresholding parameter varied. For isolation forest, the randomization seed and number of iterations is varied. For any given model, parameters or effective thresholds determine an approximate envelope in the space of TPR and FPR referred to as a Receiver Operating Characteristic (ROC) curve (Fawcett, 2006). For simplicity,

the solutions with highest accuracy (fewest false classifications) and nearest proximity to the $(0,1)$ coordinate is shown in Figure 14, alongside $T^2$ and $Q^{(X)}$ for the $\alpha = \{0.01, 0.05, 0.1\}$ significance levels. For reference, the heuristic threshold for $T^2$ from Reggente et al. (2016) is also shown.

For TOR OC, classification performance using residuals ($E$) is slightly but consistently better than than using LVs ($T$). The TPR ranges between 10–88% and FPR between 1–36% using $T$ and TPR ranges between 36–87% and FPR between 4–28%

using $E$. For TOR EC, the selected results are clustered together with a few exceptions; TPRs and FPRs are typically higher (56–85% and 8–38%, respectively). Regarding systematic differences between methods over parameters studied, the potential function and SVM methods can span a wide range of solutions in the ROC space that follows the arc delineated by the selected points shown (up to TPR and FPR of 100%), while all isolation forest solutions remained in close proximity to the points depicted in Figure 14. Both $T^2$ and $Q^{(X)}$ metrics with the significance levels explored are restricted to the upper left corner of

the ROC space as depicted.

The tradeoff in TPR and FPR is in part determined by what are designated as prediction outliers. The stratification of prediction errors by classification is illustrated in Figure 15. A classifier that is able to identify all samples with prediction errors greater than expected uncertainties would result in segregation by color in this figure. However, we see that the prediction outliers are only partially correlated with the absolute magnitude of prediction error (especially for TOR EC, where the py-

rolized fraction adds a variable contribution to precision error across samples), while samples labeled as spectroscopic outliers are more aligned with the latter. Furthermore, samples with the lowest prediction errors are also not flagged as outliers. That classifications are primarily correlated with magnitude of prediction errors (without consideration for precision) is not surprising, as measurements with higher uncertainties have higher possibility of divergence from predictions modeled by the FT-IR spectra. Biomass burning samples previously mentioned can be identified visually (and by spectral matching), but they are not




necessarily flagged as outliers with respect to the calibration models. This is not surprising as prediction errors for burning samples are not systematically higher, except for the few samples with highest TOR OC loadings.

Revisiting the classification problem posed by Reggente et al. (2016) and considering only the samples with highest prediction errors exceeding those of the 2011 IMPROVE test set as prediction outliers, it is possible to achieve TPR of 81% and FPR
of 12% for TOR OC, and TPR of 91% and FPR of 8% for TOR EC (both with the potential function method) as the solutions closest to $(0, 1)$ on the ROC curve. Outlier detection for TOR EC is better served by alternative methods to $T^2$ on account of the strong non-normality in the multivariate feature space (Reggente et al., 2016). For this scenario, selecting a classifier with high TPR comes at a cost of lowering the overall accuracy significantly because of the small proportion of high-error samples. For instance, moving from the max $D_M^2$ classifier of Reggente et al. (2016) to the potential function solution for TOR EC as
described above, an increase in TPR from 59% to 91% (a difference of 7 samples) accompanied by an increase in FPR from 1% to 8% (a difference of 142 samples) drops the overall accuracy from 98% to 92% on account of the large number of low-error samples that would be detected as being different. The desired criterion for the optimal classifier may depend on the purpose of classification. For the purposes of flagging suspicious samples during routine application of a calibration model, it may be desirable to select a classifier with high overall accuracy to keep the total number of FN and FP to a minimum. A conservative
classifier with higher TPR than low FPR is, however, likely to be more useful for model selection against a specific sample (Section 4.1.3).

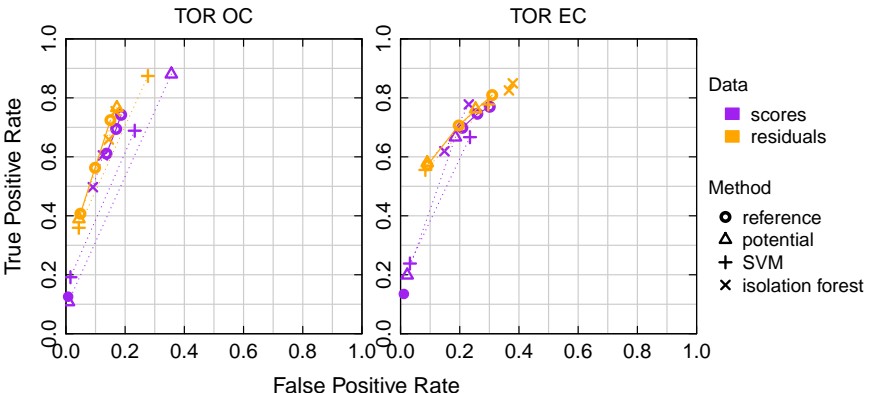

**Figure 14.** Receiver Operating Characteristic (ROC) curves for the 2013 IMPROVE data set. Symbol colors are grouped according to the data used for detection (either scores $\boldsymbol{T}$ or residuals $\boldsymbol{E}$). Symbol shapes indicate method of estimation. "Reference" denotes Hotelling's $T^2$ statistic for scores and the $Q^{(X)}$ statistic for residuals, for which three open circles are shown for the $\alpha = \{0.1, 0.05, 0.01\}$ significance levels. The filled purple symbol indicates the performance determined by the maximum $T^2$ of the 2011 IMPROVE test set, as originally used by Reggente et al. (2016). For other methods, two symbols are drawn and connected by dotted lines to indicate the solution with highest accuracy (fraction classified correctly) and the solution which lies closely to the coordinate $(0, 1)$.



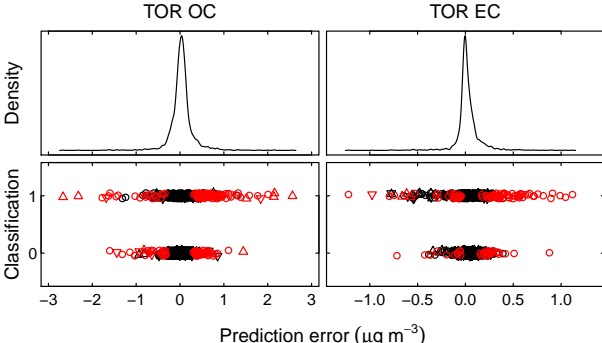

**Figure 15.** Prediction error distribution (top row) and classification results using the $Q^{(X)}$ classifier with $\alpha = 0.05$ significance level applied to model residuals (bottom row) for the 2013 IMPROVE data set. 1 corresponds to outliers and 0 as those not classified as outliers. Triangles and red samples correspond to same sample specification as Figure 12; rest of the individual prediction errors are symbolized with open circles.

### 4.1.3 Model selection

Methods for error anticipation may also be used for evaluating among a set of candidate models when reference measurements are not available to provide a full evaluation. To illustrate such an application, we revisit the apparent increase in mean prediction error shown for decreasing number of ambient samples in the calibration set displayed in Figure 9. The corresponding

increase in mean squared Mahalanobis distance between the fixed set of 253 test set spectra and those of the changing calibration set is shown in Figure 16. As $D_M^2$ increases linearly with the number of components, only the first 10 LVs are considered in each model for the purpose of a fair comparison. This example provides indication that the loss in representativeness of composition or concentration between the 253 predicted samples and calibration samples as the latter numbers are diminished (Figure 10) is reflected in the FT-IR spectra, and can be appropriately extracted after projecting them onto factor scores of their

respective PLS models.

While we have demonstrated use of $D_M^2$ to provide a qualitative comparison among several models, in principle it would be possible to use the classifiers introduced in Section 4.1.2 to find a set of models for which a new sample is not determined to be dissimilar. As mentioned in Section 4.1.2, a conservative classifier with higher TPR than low FPR is likely to be more useful for model selection for any specific sample. A sample-specific calibration model in which individual compounds from an available

database for each new prediction sample is in principle possible using concepts described in this section. However, without a priori knowledge, the most relevant features and measure of similarity among individual samples is necessarily defined through the process of calibrating a model. Therefore, it is at present time necessary to hypothesize or propose several candidate models and select among them for any new prediction sample or set of samples for possible improvements in prediction.





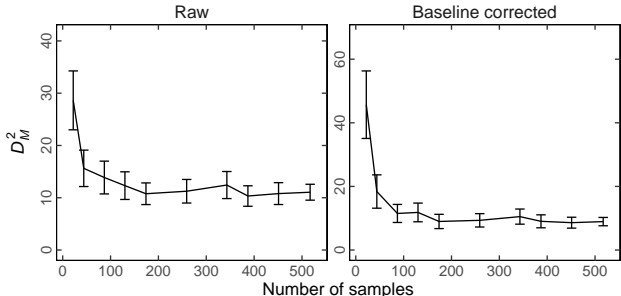

**Figure 16.** Mean squared Mahalanobis distance ($D_M^2$) between spectra of the fixed test set and changing calibration set, constructed as described in caption of Figure 9). Error bars span $\pm$ one standard deviation. The first 10 latent variables are used for estimation of $D_M^2$ in this example to reduce the dimensionality the factor space (Brereton and Lloyd, 2016).

## 4.2    Calibration maintenance

Calibration maintenance and transfer learning addresses the problem of updating a calibration model developed under one set of conditions to continue providing accurate predictions for samples measured under new conditions (Feudale et al., 2002; Torrey and Shavlik, 2009; Pan and Yang, 2010; Wise and Roginski, 2015). This topic has not yet been addressed for TOR

OC and EC calibrations using FT-IR, but we can nonetheless make a few remarks for future research needs. Difference in sampled or measured conditions can arise from changes in hardware, changes in (PTFE filter) substrate, or atmospheric aerosol composition, and imply a possible difference introduced into distributions between training and prediction data in the feature space of the model. During the operational phase of the calibration, it is therefore necessary to continuously monitor model performance and appropriateness for new samples using protocols described in Section 3.2 and Section 4.1. Notable changes

may be registered by trends in the magnitude of prediction errors compared against available reference measurements, or increasing instances of spectral outliers. The role of hardware performance in these changes can be assessed separately using the analytical protocols summarized in Section 2.3 — specifically, through the repeated analysis of laboratory check standards.

The strategy for model updating can be different according to the cause and nature of the change, but a basic premise is that the original condition still holds useful information that can be transferred to the new condition such that an entirely new

calibration is not warranted. In this way, a significant investment of resources required by model building (consisting of data collection and evaluation) may be avoided. For changes in instrument performance or installation of a separate spectrometer, commonly applied modifications range from simple linear corrections of predictions to calibration transfer algorithms to convert spectra to resemble that which may have been acquired from the primary instrument in its original state so that the original model remains applicable (Wise and Roginski, 2015; Chen et al., 2016b; Malli et al., 2017). The contribution from

PTFE can presumably be removed with the appropriate baseline correction technique (Section 3.3.2). Though not been tested extensively across various filter types, successful prediction has been reported between two PTFE filter types (Weakley et al., 2018b). Treating the PTFE signal as an interferent, training the model with additional blank (zero-analyte) samples may be an





effective approach (Ottaway et al., 2012; Kalivas, 2012; Wise and Roginski, 2015), though also requires evaluation. Changing atmospheric composition can be addressed by updating the calibration set with new samples which contain new analytes or different regimes in concentration. While there are recursive algorithms for online updating (reweighting) of models with new samples (Hayes, 1996; Helland et al., 1992; Qin, 1998; Binfeng and Haibo, 2015; Ma et al., 2015; Chen et al., 2016b), recali-

bration with the appropriate proportion of old and new samples will recreate a feature space that accommodates both groups of samples. When new samples are needed, active learning strategies seek the potentially most informative samples and minimize the requirement of new calibration samples (Douak et al., 2012).

Additional strategies from *transductive learning* aim to avoid the requirement of obtaining new samples for recalibration, but rather search for common feature representations between calibration and prediction set ("unlabeled") samples (Chapelle

et al., 2010). While these methods are more typically based on non-PLS based algorithms and applied to classification problems (Zadrozny, 2004; Cortes et al., 2005; Arnold et al., 2007; Bickel et al., 2007), some results in multivariate calibration tasks give an indication of their applicability. One approach is to reattribute weights in calibration samples to have the closest feature distribution to new samples (Huang et al., 2006; Sugiyama et al., 2008; Kim et al., 2011; Hazama and Kano, 2015; Zhang et al., 2017). New estimates weighted by their uncertainty can be furthermore be used for re-estimation of model parameters in an

iterative fashion (Culp and Michailidis, 2008; Marcou et al., 2017). Another approach is to re-estimate a feature representation in which the calibration and prediction samples are in closer proximity in this space (Culp and Michailidis, 2008; Gujral et al., 2011; Pan et al., 2011). Limited studies with PLS regression report mixed results regarding the value of incorporating unlabeled data into the calibration over simply using the original model (Culp and Michailidis, 2008; Gujral et al., 2011; Paiva et al., 2012; Bao et al., 2015). The benefit of such efforts not surprisingly depend on both the specific characteristics of the calibration

model and unlabeled data (Culp and Michailidis, 2008).

Reggente et al. (2016) showed that a recalibration strategy can improve predictions for new types of samples for the IM-PROVE network. TOR predictions for samples collected in 2013 from FRES and BYIS sites had not only high instances of prediction errors, but also systematic biases when using the 2011 IMPROVE model. A dedicated calibration model built with two-thirds of the available data set at the two new sites improved prediction performance for samples reserved for testing (Ta-

ble 3). Whether to incorporate new types of samples into the original calibration set to build a monolithic model, or to unify the calibrations through a multilevel modeling framework may depend on the number and leverage of new samples. A model derived from including new samples with old may cease to perform adequately for the original types of samples. From a case study in 2013 CSN (Weakley et al., 2018a), including ELLA samples in the calibration did not seem to affect the non-ELLA samples, but ELLA samples were also found to have not have much leverage within the scope of all samples. When updating

an existing model, it is necessary to re-evaluate the model for old as well as new types of samples.

## 5 Conclusions

The FT-IR spectra of PM is rich in chemical information, and quantitative information such as TOR-equivalent OC and EC can be extracted from it provided that we can find the appropriate combination of training samples and algorithms for extraction.



**Table 3.** Figures of merit for selected FRES (Fresno, CA) and BYIS (Baengnyeong Island, S. Korea) samples using base case 2011 IMPROVE calibration and a "dedicated" model built only using samples from FRES and BYIS.

| Model | Variable | Samples | Bias ($\mu g\,m^{-3}$) | Error ($\mu g\,m^{-3}$) | $R^2$ |
|-------|----------|---------|------|-------|-------|
| 2011 IMPROVE | OC | FRES, BYIS | 0.28 | 0.43 | 0.79 |
| Dedicated | OC | FRES, BYIS | -0.03 | 0.16 | 0.96 |
| 2011 IMPROVE | EC | FRES | 0.05 | 0.10 | 0.85 |
| Dedicated | EC | FRES | 0 | 0.06 | 0.93 |
| 2011 IMPROVE | EC | BYIS | 0.13 | 0.17 | 0.60 |
| Dedicated | EC | BYIS | -0.07 | 0.11 | 0.66 / 0.84[*] |

[*] one outlier removed.

In this manuscript, we review procedures for spectral processing and data-driven calibration, where the data are taken from collocated measurements of TOR OC and EC. In this effort, procedures for initial steps for model building and evaluation, and later steps for monitoring of model behavior during the operational phase of a calibration model are described.

The number and types of samples required for calibration is determined by the diversity of composition in the prediction set. When samples are selected from the same sites as the prediction set, FT-IR calibration models could predict with virtually no bias and errors within 0.15 $\mu g\,m^{-3}$ for TOR OC and 0.11 $\mu g\,m^{-3}$ for TOR EC for areal loadings in the 2011 IMPROVE and 2013 CSN networks. Less than 5% of samples fell below the estimated detection limit. These metrics are on a par with the reference measurement evaluated for the same year. For the 2011 IMPROVE data set, the number of ambient calibration samples can be reduced from the canonical number of 501 down to approximately 150 samples and maintain similar prediction performance for the diversity in composition represented by 237 samples. To the extent that we have experimented (virtually) for TOR OC, the limitation is likely due to the difficulty in maintaining the same distribution of ammonium to OC ratio in the calibration set as in the test set with fewer number of samples obtained by the temporal and spatial stratified sample reduction approach illustrated.

As evaluated for the IMPROVE network, TOR-equivalent concentrations in new samples collected for a later year (2013) and more sites (11 additional ones) have similar performance metrics overall, with exception to samples from two new sites (FRES and BYIS) not in the calibration set. Higher prediction errors for TOR OC occur largely due to specific types of samples not well-represented in the calibration year. While these samples are predicted without bias, their errors are higher on account of the higher areal loadings of TOR OC beyond the range of original calibration. Estimates of prediction intervals for both TOR and model predictions suggest that more than 92% of samples are predicted within anticipated precision errors. Outlier detection methods can be used to detect samples which are different with respect to the modeled domain to provide some indication of the magnitude of prediction errors. However, accurate detection of high-error samples comes with a tradeoff of increased false positive rates; the outlier detection method can be selected based on the application and desired tolerance for each type of detection error (false positive or negative). An obvious solution for reducing prediction errors in different samples




is to acquire new samples for recalibration, though judicious calibration maintenance strategies (e.g., sample reweighting) can potentially minimize the number of new samples needed.

The procedure for quantitative prediction of TOR-equivalent OC and EC is a statistical one and depends the ability of an algorithm to resolve the overlapping absorption bands in the mid-IR and relate relevant features to the concentration of the target analyte. Given the evolving diversity in aerosol composition, it is not clear that arriving at an invariant, universal calibration model applicable for every new sample is practical. However, in describing the broader context of chemometrics and machine learning algorithms that are available for addressing each stage of the model life cycle, challenges for calibrating complex spectra are not insurmountable provided that they are systematically handled as described in this paper. We can use a wide range of statistical quality control procedures at our disposal to assess similarity of relevant features among spectra to continually monitor model performance, to anticipate appropriateness of existing calibration models, and to propose revisions. Construction of calibration models specific to individual or groups of samples may be envisioned provided that we are further able to identify the most important spectral features to assess similarities relevant for TOR OC and EC estimation.

In parallel to ensuring numerical accuracy of a calibration, understanding how the calibration relates spectral absorbances to TOR concentrations is critical for anticipating model applicability. Identification of important vibrational modes used in the calibration facilitates understanding of how the model relates absorbances to concentrations of the target analyte. Moreover, this association can be used to gain a better understanding of molecular structure in complex substances underlying the OC and EC concentrations reported by TOR. For TOR OC, functional groups typically associated with atmospheric organic matter were found: aliphatic CH, carbonyls, and nitrogenated functional groups. For TOR-equivalent EC prediction, the vibrational mode associated with C-C stretch of aromatic rings typically observed in mid-IR spectra of soot appears to be an important absorption band, but a model for Elizabeth, NJ, was able to predict TOR-equivalent EC concentrations accurately without use of this spectroscopic region. While attempts to understand model LVs have thus far been limited, some work by Weakley et al. (2016) indicate that 2013 CSN aerosols could be modeled with a surprisingly few LVs, with nearly 90% of the variation in TOR OC explained by one variable. Further analysis of constituent samples using source apportionment techniques and analysis of chemical composition (e.g., using functional groups) are bound to benefit overall model interpretation.

In summary, this manuscript outlines a general perspective and specific practices for model building; encompassing judicious specification of algorithm, spectra processing procedure, and sample selection. Taking a systematic approach toward calibration with a diverse set of reference measurements allows us to expand the suite of information extractable from FT-IR spectra, to complement functional group analysis from laboratory calibrations, which has long been the focus. Given the demonstrated simplicity and non-destructive nature of acquiring spectra from PTFE filters, this technique can expand TOR-equivalent OC and EC measurements (which has a long history) to new campaigns and new locations in which only PTFE samples are collected for gravimetric reference measurements. Therefore, we anticipate that the procedure outlined in this paper can complement existing methods for PM monitoring with TOR-equivalent OC and EC, and provide guidance in extracting composition of substances from FT-IR spectra of atmospheric PM. Given that a wide range of inorganic and organic substances display mid-IR activity, further exploration of data sources and algorithms for quantitative analysis can continue to expand the cost-effective application of FT-IR in chemical speciation measurements.



*Code availability.* Companion paper and web platform http://airspec.epfl.ch

*Data availability.* The IMPROVE and CSN network data will be made publically available.

*Competing interests.* The authors have no competing interests.

*Disclaimer.* None.

5 *Acknowledgements.* The authors acknowledge funding from EPFL, Swiss National Science Foundation (200021_143298, 200021_169506), and the U.S. EPA and the IMPROVE program (National Park Service cooperative agreement P11AC91045). We also thank the IMPROVE team at UC Davis for performing the sample handling and site maintenance for all IMPROVE sites and the RTI International team for managing the CSN during the 2013 sampling year.



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
