# Peer review of "Atmospheric particulate matter characterization by Fourier Transform Infrared spectroscopy: a review of statistical calibration strategies for carbonaceous aerosol quantification in US measurement networks"

_Atmospheric Measurement Techniques, 2018_

## Referee Comment (RC1) · Anonymous Referee #1 · 6 Aug 2018

General comments: "Atmospheric particulate matter characterization by Fourier Transform Infrared spectroscopy: a review of statistical calibration strategies for carbonaceous aerosol quantification in US measurement networks" addresses the need for accurate calibration models to interpret Fourier Transform Infrared (FT-IR) spectra, which can provide quantification of multiple species in complex atmospheric particulate matter (PM) mixtures. This review focuses on using models to facilitate quantitative predictions of organic carbon (OC) and elemental carbon (EC). The manuscript contributes

substantial data thoroughly exploring several quantitative models for interpreting FT-IR spectra. The approaches and methods are valid and balanced, and maintain consistent self-checks for validity and bias.

Specific comments: 1. The scope of the manuscript is very ambitious and covers detailed ground ranging from sample collection to model calibration maintenance. Potentially, the paper could be divided into two manuscripts. The manuscript starting from Background to the end of Section 3 could be its own review paper of methods and data-driven model-building, and Section 4 might be another review paper on operational phase models calibration and error prediction. 2. The manuscript might benefit from a table that presents/overviews all the components explored in model building, evaluation, and interpretation. 3. The figures in general can be made larger to read the text more clearly. In particular, Figures 7 and 13 have very small text. Additionally, symbols can be used in Figure 10 to ensure readability in greyscale.

Technical corrections: 1. Page 9 line 23 manufacturer of the PTFE filters should probably say "Whatman". 2. Page 19 line 3 abbreviation is "SAFOX" but Figure 6 uses the abbreviation "SAFO". 3. Page 23 line 9 should say, "A formal comparison... has not been performed, ...". 4. Page 24 line 12 should probably say "Wrappers operate under the implicit assumption...". 5. Page 28 line 26 should probably say "While the large number of LVs used by the IMPROVE calibration models precluded attempts at identifying individual components..."

---

## Referee Comment (RC2) · Anonymous Referee #3 · 29 Aug 2018

General Comments:

Takahama et al (2018), hereafter referred to as T2018 reviews methods used to determine quantitative measurements of OC/EC using FTIR spectroscopy on PTFE filter samples taken from sampling networks such as the Chemical Speciation Network (CSN) and the Interagency Monitoring of Protected Visual Environments (IMPROVE) network. The topic and scope of the article are very appropriate for AMT. There have

been many papers on this topic in the past few years, so a review article summarizing the calibration, processing, and evaluation techniques is timely. However, this paper does not stand alone. It requires knowledge of the authors' previous papers, as well as other information. There are too many gaps in the information provided. Examples include such as why PTFE is used and not quartz fiber filters, the lack of general references for the measurement networks, and the lack of legends on the plots. Additionally, there are some organizational problems, ranging from unclear section titles, to different datasets used in different ways, requiring effort from the reader to keep everything straight. Stated differently, this paper seems to focus on what the authors do and how they do it, which is important, but the authors need to emphasize why this is important, how it fits within the larger body of relevant literature, and make the paper stand on its own, while being clear to read. Thus, while this paper definitely should be published in AMT, it requires major revisions to improve the clarity of writing.

Specific Comments:

I can't list all the areas where the authors could improve the organization, only a few are mentioned here.

Site selection and use: It's not always clear what datasets are used when, the map on Figure 1 suggests that the BYIS and FRES IMPROVE sites are used only as testing samples, but at the end of the paper the BYIS and FRES sites are used for training a new predictor. The 'data' section mentions calibration data from IMPROVE 2011, and CSN 2013, but it isn't clear that each of these datasets will be used separately and at different points within the paper. Thus, at various points in the paper: 2011 IMPROVE data are used to test calibrations using 2011 IMPROVE data, 2013 IMPROVE measurements are used to test a calibrations with 2011 IMPROVE data, 2013 IMPROVE data (For FRES and BYIS) are used to test calibrations with 2013 IMPROVE data, and 2013 CSN data are used to test calibrations with 2013 CSN data. In a paper this long, it can be challenging to remember what data is used where.

Some other site selection questions: A user might wonder why CSN and IMPROVE data are not used together to develop a model: there are some differences in the collection method between CSN and IMPROVE (Weakley et al., 2016), but no discussion of these differences is found in the paper. The Elisabeth (ELLA) required a special calibration. Weakley et al. (2016) noted that this site was located near a refinery but no discussion of the potential influence of the refinery on different spectra is discussed.

Section titles could be improved. Section 4.2 is an example of this, a more descriptive title like "Applicability of calibrations developed under one set of conditions to samples measured under new conditions" would be more descriptive.

One of of the use cases for this technique is in a network where OC/EC measurements are not made using the standard EGA technique (Page 6, Line 30). The discussion in 4.2 directly touches this concept - but the topic mentioned in the introduction isn't really brought up in that section.

The data availability does not mention where to obtain the FTIR Spectra. Also, I could not find the source code at http://airspec.epfl.ch.

The authors should include a list of acronyms as an appendix.

There are no calibration sites in the Midwestern states (e.g. at longitudes between Birmingham, Alabama and Mesa Verde, CO), other than the Sac and Fox site which only has one half years' worth of data. Is this a problem for applicability of the model?

The authors briefly mention meteorological influences on the calibrations – could model error be used to infer something about the variability due to meteorological conditions? Are there better performances across the different months – e.g. both meteorological patterns, as well as well as combustion patterns, are different between summer and winter. How does calibration data during only a short period (such as at the Sac and Fox site) bias the results?

Specific Minor Comments:

Page 3, Line 9 is written as P3L9 P3L9 – needs a citation.

P9L15: The authors should cite Malm and Hand (2007) or Solomon et al. (2014).

P9L21: The authors should cite (Solomon et al., 2014)

P10L1: Are the two sites that were collocated from IMPROVE, or was one IMPROVE site collocated with one CSN site?

P20L1: The Situation with the ELLA predictions is problematic – how many locations will need specific trainings associated with them? Probably should mention that the Elizabeth site is located next to a refinery (Weakley et al., 2016).

P32L31: Samples from these sites – is this for 2011 or 2013?

P38L31 – NOx isn't a surrogate for EC – but it may give an indication or be useful for prediction of EC. Somewhere – should mention that PTFE filters should have lower gas-phase adsorption than quartz fiber filters (Turpin et al 1994, Gliardoni et al 2007). The BYIS site is very interesting as it is on another continent – this is an interesting use case, and basically is an example of using a model trained in one location or with a network to apply to another location – and as noted, this method does not do well under that scenario.

P45L20: Also, Figure 1 indicates that BYIS is only used as a test dataset, not calibration dataset. Is this correct? So then wouldn't we expect some errors in BYIS since that location is on a different continent than the trainings? What is similar about FRES and BYIS? What about South Korea – it's very interesting that this location is not an outlier, given that the models were trained on an entirely different continent.

Table 3: Why are OC combined for FRES and BYIS, but separated out for EC?

Figure 1: It would be helpful to also include the site abbreviation, either in the location labels in this figure, or in a table. The authors identify the abbreviations the first time they are used, which is good, but the paper is long enough I found myself having to

refer back.

Figure 12 - The authors need to clarify in the paper what the different meanings of the two types of triangles - I assume it is just meant to show two different similar types of spectra, associated with wildfires. For the wildfire cases, are there significant differences between the red spectra (those with predicted-observed difference > combined uncertainties) and the black spectra? Finally, the subplots should be labeled a,b,....f, and there should be a legend identifying black, red, and the two different triangle containing spectra. It might be good to mention the importance of aliphatic stretches and carbonyl vibrations.

Figure 14 – add the filled circle to the legend.

Figure 15 – Similar comments to Figure 15 – please restate the descriptions of triangles and red samples, and include a legend in the plot. Also, subplot labels will help with interpretation.

Technical Corrections:

P3L25: smog chamber -> smog chambers

P6L23: measured by EGA EC is an -> measured by EGA, EC is an

P6L25: and therefore less influenced -> and therefore is less influenced

P9L31: "Change to artifact correction method for OC carbon fractions" – not sure what this is.

P20L5: calibrations models -> calibration models

P30L6: for new smaples. -> for new samples

P36L10: for the rediction standard error -> for the prediction standard error

References: There were many errors in the references section, and it would take too much time for me find and fix all of the problems. A non-exhaustive list follows, with

some examples. Please redo the references section.

- Many of the references appear to be missing journals, and then the title ends up being formatted like the journal. Examples include Cunningham et al (1976), Efron and Tibshirani (1996).

- The Debus et al (2018) citation has no information other than author, title, and year, and the fact that it was accepted in some journal. This is a problem because section 2.3 cites Debus et al (2018) heavily. I tried to find it by searching online but could not.

- Book and grey literature references are incorrect, e.g. Mahalanobis, P "On the generalised distance in statistics", Tibshirani (2014)

References: Malm, W. C., and Hand, J. L.: An examination of the physical and optical properties of aerosols collected in the IMPROVE program, Atmospheric Environment, 41, 3407-3427, https://doi.org/10.1016/j.atmosenv.2006.12.012, 2007.

Solomon, P. A., Crumpler, D., Flanagan, J. B., Jayanty, R. K. M., Rickman, E. E., and McDade, C. E.: U.S. National PM2.5 Chemical Speciation Monitoring Network-s—CSN and IMPROVE: Description of networks, Journal of the Air & Waste Management Association, 64, 1410-1438, 10.1080/10962247.2014.956904, 2014.

Weakley, A. T., Takahama, S., and Dillner, A. M.: Ambient aerosol composition by infrared spectroscopy and partial least-squares in the chemical speciation network: Organic carbon with functional group identification, Aerosol Science and Technology, 50, 1096-1114, 10.1080/02786826.2016.1217389, 2016.

---

## Author Comment (AC1) · 11 Nov 2018

**Response to Reviewers**

We thank the editor and anonymous reviewers for taking the time to oversee this process. Our point-by-point responses are included below in blue, and changes to the manuscript are highlighted in red.

**Reviewer 1**

General comments: "Atmospheric particulate matter characterization by Fourier Transform Infrared spectroscopy: a review of statistical calibration strategies for carbonaceous aerosol quantification in US measurement networks" addresses the need for accurate calibration models to interpret Fourier Transform Infrared (FT-IR) spectra, which can provide quantification of multiple species in complex atmospheric particulate matter (PM) mixtures. This review focuses on using models to facilitate quantitative predictions of organic carbon (OC) and elemental carbon (EC). The manuscript contributes substantial data thoroughly exploring several quantitative models for interpreting FT-IR spectra. The approaches and methods are valid and balanced, and maintain consistent self-checks for validity and bias.

We thank the reviewer for the assessment.

Specific comments:

1. The scope of the manuscript is very ambitious and covers detailed ground ranging from sample collection to model calibration maintenance. Potentially, the paper could be divided into two manuscripts. The manuscript starting from Background to the end of Section 3 could be its own review paper of methods and data-driven model-building, and Section 4 might be another review paper on operational phase models calibration and error prediction.

   We agree with the reviewer that this paper covers a lot of ground, but a large part of its value (currently lacking in the literature) is in bringing together different aspects of the calibration roadmap in a single source, and illustrating it with an extended example that is used across all parts.

2. The manuscript might benefit from a table that presents/overviews all the components explored in model building, evaluation, and interpretation.

   We have included a concise overview in Table C1.

3. The figures in general can be made larger to read the text more clearly. In particular, Figures 7 and 13 have very small text. Additionally, symbols can be used in Figure 10 to ensure readability in greyscale.

   We thank the reviewer for this comment. We have increased the font size in Figures 7 and 13 and changed the symbols in Figure 10 (and also Figure 3).

Technical corrections:

1. Page 9 line 23 manufacturer of the PTFE filters should probably say "Whatman". corrected

2. Page 19 line 3 abbreviation is "SAFOX" but Figure 6 uses the abbreviation "SAFO". should be SAFO

3. Page 23 line 9 should say, "A formal comparison. . . has not been performed, . . ." corrected

4. Page 24 line 12 should probably say "Wrappers operate under the implicit assumption. . ." corrected

5. Page 28 line 26 should probably say "While the large number of LVs used by the IMPROVE calibration models precluded attempts at identifying individual components. . ." corrected

We thank the reviewer for catching these errors.

Additional comments from Quick Review Report

1. Section 4.1 [operational phase] is not clearly connected back to Sections 3.4 and 3.5 (model interpretation and sample selection). For example, Section 3.4 points the reader to Section 4.1.2 [outlier detection] in particular, warning of the detriment in applying models in different contexts. However, Section 4.1.2 never again mentions the relationship between its content (outlier detection) and its impact on Section 3.4 (model interpretation). Additionally, while Section 4.1.2 Outlier detection makes a connection back to Section 3.5 Sample selection, it does not make the connection back to its impact on model interpretation. Perhaps when asking the reader in Section 3 to see information in Section 4, it can be restated in the beginning of Section 4 (or the appropriate subsection) why the reader might be interested in doing so.

We thank the reviewer for this comment. We have made the connection more clear by adding the following text to Section 4 [operational phase]:

"[T]his is the eventual use case for such calibration models — for instance, to enable FT-IR to provide TOR-equivalent carbon values from a PTFE filter in new monitoring sites or measurement campaigns where TOR analysis from a separate filter is not available. Without reference measurements, it is important to evaluate the appropriateness of available calibration models for new samples, continually monitor the performance of the model by introspective means, and update the calibration as necessary."

Text in Section 4 [operational phase] now further ties back to Section 3.4 and 3.5 (new additions in italic):

"Without reference measurements, many external indicators might be used to characterize differences between new samples and those in the calibration set, *especially with respect to attributes identified to be important (Section 3.5.1).*

[...]

"Spectral matching combined with model interpretation (Section 3.4) can identify particular sample types that may be problematic for a calibration model *a priori*."

and in Section 3.4 [interpretation] is changed to (new text in italic):

"In particular, it is possible to exploit statistical correlations among the variables to make predictions, which can be detrimental if the correlation changes or model is applied in a different context . Therefore, model interpretation is strongly related to anticipation of model applicability *and a priori identification of samples with potentially high prediction errors (Section 4.1.2).*"

2. The introduction of the manuscript is inadvertently misleading in how much it reviews laboratory-generated mixtures and simulated spectra. The introduction spends 2–3 (out of 4) pages reviewing methods and shortcomings for laboratory-generated mixtures and simulated spectra, which gives the impression that the bulk of the manuscript may also review laboratory standards and synthetic spectra. Perhaps the introduction can include subsection headers, such as "Limits of early applications" and "Data-driven approach."

We thank the reviewer for the suggestion. We have added subsection headings, though we have named them "limits of conventional approaches to calibration" (Section 1.1) and "using collocated measurements" (Section 1.2) as some methods using laboratory standards can be considered to be data-driven, and are also currently in use. The use of collocated measurements provides a complementary approach, which we have now stated in the Introduction:

"[U]se of collocated measurements complement conventional approaches in expanding the capabilities of FT-IR spectroscopy to extract useful information contained in vibrational spectra."

**Reviewer 2**

General comments:

Takahama et al (2018), hereafter referred to as T2018 reviews methods used to determine quantitative measurements of OC/EC using FTIR spectroscopy on PTFE filter samples taken from sampling networks such as the Chemical Speciation Network (CSN) and the Interagency Monitoring of Protected Visual Environments (IMPROVE) network. The topic and scope of the article are very appropriate for AMT. There have been many papers on this topic in the past few years, so a review article summarizing the calibration, processing, and evaluation techniques is timely.

We thank the reviewer for the assessment.

1. However, it requires knowledge of the authors' previous papers, as well as other information. There are too many gaps in the information provided. Examples include such as

   - why PTFE is used and not quartz fiber filters

     We have added the following statements (in italics) in Section 2.1 [background on FT-IR]:

     "In this section, we cover the background necessary to understand FT-IR spectroscopy in the analysis of PM collected onto PTFE filter media, *which is optically thin and permits an absorbance spectrum to be obtained by transmission without additional sample preparation (McClenny et al., 1985; Maria et al., 2003).*"

     and in Section 2.2 [background on sample collection]:

     "*PTFE filters are used for gravimetric analysis on account of its low vapor absorption (especially water) and standardization in compliance monitoring, while quartz fiber filters are separately collected on account of its thermal stability (Chow, 1995; Chow et al., 2007b; Malm et al., 2011; Solomon et al., 2014; Chow et al., 2015).*"

   - the lack of general references for the measurement networks

     We have added references to Malm and Hand (2007) (IMPROVE) and Solomon et al. (2014) (CSN).

   - the lack of legends on the plots.

     We have included description of symbols and colors in the caption for space-constrained figures (which is a more traditional convention). We have modified the caption of Figure 7 to be more descriptive.

2. Additionally, there are some organizational problems, ranging from unclear section titles, to different datasets used in different ways, requiring effort from the reader to keep everything straight.

   We have renamed section titles to be more informative. For instance, "Interpretation" has been renamed to "Interpretation of important variables and their interrelationships" and "Model selection" to "Model selection without reference measurements" and "Calibration maintenance" to "Updating the calibration model". We have additionally clarified how the data is used; further clarification and corresponding changes to the paper are included in responses to specific comments 1–3.

3. Stated differently, this paper seems to focus on what the authors do and how they do it, which is important, but the authors need to emphasize why this is important, how it fits within the larger body of relevant literature, and make the paper stand on its own, while being clear to read.

We thank the reviewer for this perspective. In constructing this paper around a roadmap for statistical calibration, the importance of this paper is to provide generalization beyond what has been accomplished thus far for predicting TOR-equivalent carbon measurements and to enable extraction of useful information from vibrational spectra at an operational scale. This "in-situ" approach to calibration — using collocated measurements as reference — is an emerging strategy in atmospheric measurement research (e.g., for low-cost sensor calibration development) to increase the number of available measurements beyond that which more expensive reference measurements can provide. For the use case of FT-IR measurement of $PM_{2.5}$, the review pervasively draws upon the work of the authors in carbon quantification as an extensive example, as none other exist to our knowledge; a large number of citations are made with respect to the greater body of relevant literature in statistical calibration and statistical process control from which this framework is developed.

We have added restructured part of the introduction regarding calibration with collocated measurements and our demonstrated application to TOR analysis such that it reads:

"The benefit of building data-driven calibration models to reproduce concentrations reported by available measurements is twofold. One is to provide equivalent measurements when the reference measurements are expensive or difficult to obtain. For example, FT-IR spectra can be acquired rapidly, non-destructively, and at low cost from from Polytetrafluoroethylene (PTFE) filters commonly used for gravimetric mass analysis in compliance monitoring and health studies. That vibrational spectra contain many signatures of chemical constituents of PM (which also gives rise to challenges in spectroscopic interpretation) provides the basis for quantitative calibration of a multitude of substances. This capability for multi-analyte analysis is beneficial when a single filter may be relied upon during short-term campaigns, or in network sites for which installation of the full suite of instruments is prohibitive. The second benefit is the ability to gain a better understanding of atmospheric constituents measured by other techniques by associating them with important vibrational modes structural elements of molecules identified in the FT-IR calibration model. Such an application can be enlightening for studying aggregated metrics such as carbon content, or functional group composition in atmospheric PM quantified by techniques requiring more sample mass and user labor: ultraviolet-visible spectrometry or nuclear magnetic resonance spectroscopy (Decesari et al., 2003; Ranney and Ziemann, 2016).

In this paper, we demonstrate an extensive application of this approach in the statistical calibration of FT-IR spectra to collocated measurements of carbonaceous aerosol content — organic carbon (OC) and elemental carbon (EC) — characterized by a particular type of evolved gas analysis (EGA). EGA includes thermal optical reflectance (TOR) and thermal optical transmittance (TOT), which apportions total carbon into OC and EC fractions according to different criteria applied to the changing optical properties of the filter under stepwise heating (Chow et al., 2007a). EGA OC and EC are widely-measured in monitoring networks (Chow et al., 2007a; Brown et al., 2017), with historical significance in regulatory monitoring, source apportionment, and epidemiological studies. While EC is formally defined as $sp^2$-bonded carbon bonded only to other carbon atoms, what is measured by EGA EC is an operationally-defined quantity which is likely associated with low-volatility organic compounds (Chow et al., 2004; Petzold et al., 2013; Lack et al., 2014). EGA OC comprises a larger fraction of the total carbon and therefore less influenced by pyrolysis artifacts that affects quantification of EGA EC. In addition to OC estimates independently constructed from laboratory calibrations of functional groups, prediction of EGA OC and EC from FT-IR spectra will provide values for which strong precedent in atmospheric studies exist. Thus, use of collocated measurements complement conventional approaches in expanding the capabilities of FT-IR spectroscopy to extract useful information contained in vibrational spectra."

Specific Comments:

1. Site selection and use:

   - It's not always clear what datasets are used when, the map on Figure 1 suggests that the BYIS and FRES IMPROVE sites are used only as testing samples, but at the end of the paper the BYIS and FRES sites are used for training a new predictor.

     We have modified Figure 1 and its caption, and added the following statement to Section 2.2 [background on sample collection]:

     "TOR-equivalent carbon predictions for 2011 and 2013 IMPROVE samples discussed for this paper are made with a calibration model using a subset of samples from 2011 IMPROVE, and TOR predictions for 2013 CSN samples are made with a calibration model using a subset of samples from 2013 CSN. One exception is a special model constructed to illustrate how new samples can improve model prediction (Section 4.2); a subset of samples from two sites — Fresno, CA (FRES) and Baengnyeong Island, S. Korea (BYIS) — in 2013 IMPROVE are used to make predictions for the remaining samples at those sites. In all cases, analytical figures of merit for model evaluation are calculated for samples that are not used in calibration."

   - The 'data' section mentions calibration data from IMPROVE 2011, and CSN 2013, but it isn't clear that each of these datasets will be used separately and at different points within the paper. Thus, at various points in the paper: 2011 IMPROVE data are used to test calibrations using 2011 IMPROVE data, 2013 IMPROVE measurements are used to test a calibrations with 2011 IMPROVE data, 2013 IMPROVE data (For FRES and BYIS) are used to test calibrations with 2013 IMPROVE data, and 2013 CSN data are used to test calibrations with 2013 CSN data. In a paper this long, it can be challenging to remember what data is used where.

     The modification to the text in response to the point above also now explicitly states that subsets of 2011 IMPROVE and 2013 CSN data are used to build calibration models for IMPROVE and CSN predictions, respectively, with the exception of the special model using FRES and BYIS samples.

2. Some other site selection questions: A user might wonder why CSN and IMPROVE data are not used together to develop a model: there are some differences in the collection method between CSN and IMPROVE (Weakley et al., 2016), but no discussion of these differences is found in the paper. The Elisabeth (ELLA) required a special calibration. Weakley et al. (2016) noted that this site was located near a refinery but no discussion of the potential influence of the refinery on different spectra is discussed.

   We thank the reviewer for pointing out the potential to use IMPROVE and CSN data together for calibration. We have now addressed this point by adding the following statement in Section 2.2 [background on sample collection]:

   "Given the different sampling protocols that result in different spectroscopic interferences from PTFE (due to different filter types) and range of mass loadings (due to flowrates), and difference in expected chemical composition (due to site types), calibrations for the CSN and IMPROVE networks have been developed separately (Weakley et al., 2016). Advantages of building such specialized models in favor of larger, all-inclusive models are discussed in Section 3.5."

   Regarding Elizabeth, NJ, (ELLA) the site was located near a toll station in the NJ turnpike (not refinery) and the impact of potentially high levels of diesel PM are discussed in Section 3.4, to which we added the possible impact of the nearby source (in italics).

   "Weakley et al. (2018) found that a calibration model for ELLA did not require aromatic structures for prediction of TOR-equivalent EC. This site was *located in close proximity to a toll station on the New Jersey turnpike and was* characterized by high diesel PM loading, low OC/EC ratio, and low degree of

charring compared to samples from other CSN sites in the 2013 data set. The calibration model was able to predict TOR- equivalent EC concentrations primarily using absorption bands associated with aliphatic C-H (also selected in the calibration model for the other 2013 CSN sites) and nitrogenated groups believed to be markers for diesel PM."

3. Section titles could be improved. Section 4.2 is an example of this, a more descriptive title like "Applicability of calibrations developed under one set of conditions to samples measured under new conditions" would be more descriptive.

   Calibration maintenance is a technical phrase used in chemometrics but we have renamed the section to "Updating the calibration model" to be more descriptive; other changes to section titles are included in response to general comment 2.

4. One of the use cases for this technique is in a network where OC/EC measurements are not made using the standard EGA technique (Page 6, Line 30). The discussion in 4.2 [calibration maintenance] directly touches this concept - but the topic mentioned in the introduction isn't really brought up in that section.

   This is actually relevant for both Sections 4.1 [anticipating errors] and 4.2 [updating calibration]. We have changed the opening statement of Section 4 [operational phase] to include the italicized statement:

   "The operational phase of the model marks a departure from the building and evaluation phases (Figure 2) in that reference measurements may no longer be available on a regular basis. *However, this is the eventual use case for such calibration models — for instance, to enable FT-IR to provide TOR-equivalent carbon values from a PTFE filter in new monitoring sites or short-term field campaigns where collection and analysis of PM on a separate quartz fiber filter for TOR analysis is prohibitive. Without reference measurements, it is important to* continually monitor the performance of the model by introspective means, and update the calibration as necessary."

   And in Section 4.2 [updating calibration]:

   "In the context of FT-IR measurements, TOR reference measurements may not be available for short-term campaigns at new sites and some aspects of transfer learning and transductive learning strategies (sample reweighting or basis-set rederivation) may be the only option for improvement if prediction errors from existing calibration models are expected to be high (Section 4.1). For long-term operation at a fixed site, collecting a limited number of reference samples for recalibration initially or periodically can be a viable strategy if sample characteristics substantially differ from those available for calibration."

5. The data availability does not mention where to obtain the FTIR Spectra. Also, I could not find the source code at http://airspec.epfl.ch.

   The FT-IR data will be hosted in a publicly-accessible repository, but in the meantime can be obtained from the authors directly. The source code for AIRSpec is currently under review with another manuscript but can be obtained together with its underlying packages at `https://aprl.epfl.ch/page-130782-en.html`.

6. The authors should include a list of acronyms as an appendix.

   We have included Table B1 in the appendix that lists acronyms used in multiple sections.

7. There are no calibration sites in the Midwestern states (e.g. at longitudes between Birmingham, Alabama and Mesa Verde, CO), other than the Sac and Fox site which only has one half years' worth of data. Is this a problem for applicability of the model?

   The range of sites, local sources, and the meteorological conditions represented in the calibration samples are relevant only to the extent that they add to the diversity of chemical composition, which enables application of the model to new samples with similar composition. As we do not have many sites in the Midwest with which we can evaluate the model, presently it is difficult to determine whether

the current calibration models would suitable. If FT-IR spectra were available from Midwestern sites (without TOR measurements), error anticipation methods (Section 4.1) can be used together with laboratory calibrations (e.g., functional group measurements) to determine how similar the samples are spectroscopically and chemically to samples already present in the current calibration set (from mostly non-Midwestern states).

We have modified the text as included in response to comment 8 below.

8. The authors briefly mention meteorological influences on the calibrations – could model error be used to infer something about the variability due to meteorological conditions? Are there better performances across the different months – e.g. both meteorological patterns, as well as well as combustion patterns, are different between summer and winter. How does calibration data during only a short period (such as at the Sac and Fox site) bias the results?

The variability due to meteorology and other environmental conditions are only relevant to the extent that they change the sample composition outside of the range encountered in the calibration set. In Section 4 [operational phase], we summarize how Reggente et al. (2016) found that calibration developed under one year was able to provide predictions for another year (and also at different sites), but this is possibly because variations across all seasons were represented in the calibration set. Using calibration from a short period can bias predictions if the range of composition encountered over subsequent periods (e.g., different seasons) are not well-represented during the short period of calibration. This is an area that can benefit from complementary spectral analysis to assess variability (e.g., in terms of functional group composition).

We have modified Section 3.2.2 [model evaluation] with italics indicating new additions:

"For instance, high prediction errors elevated over multiple days may be associated with aerosols of *unusual* composition transported under synoptic scale meteorology *that is not well-represented in the calibration samples*."

In Section 3.5 [sample selection], regarding use of the stratified sample selection approach (selecting samples spaced out over one year at each measurement site), the following text has been added:

"[S]amples from the same site and season are not strictly required for successful prediction of each new sample. Reggente et al. (2016) demonstrate accurate prediction for a full year of TOR OC and EC concentrations at sites not included in the calibration (also revisited in Section 4.1). The extent to which site, season, local emission, or meteorological regime of a new sample affects prediction depends on how these factors contribute to deviation in chemical composition from calibration samples."

And in Section 4.1.2 [outlier detection], the italicized statement has been added:

"[T]he actual increase in prediction error (if any) will depend on the functional relationship among variables and how well they are represented by the model — e.g., a linear relationship modeled by a linear mapping may perform adequately in interpolation and extrapolation. *For instance, samples with OM/OC and OC/EC composition and TOR OC concentrations out of range with respect to calibration samples were predicted without substantial increase in errors (Section 3.5.1).* Therefore, not all outliers may be associated with high prediction errors."

Technical corrections:

1. P3L25: smog chamber → smog chambers corrected

2. P6L23: measured by EGA EC is an → measured by EGA, EC is an corrected to "EC measured by EGA"

3. P6L25: and therefore less influenced → and therefore is less influenced corrected

4. P9L31: "Change to artifact correction method for OC carbon fractions" – not sure what this is. This was a citation error

5. P20L5: calibrations models → calibration models corrected

6. P30L6: for new smaples. → for new samples corrected

7. P36L10: for the rediction standard error → for the prediction standard error corrected

References: There were many errors in the references section, and it would take too much time for me find and fix all of the problems. A non-exhaustive list follows, with some examples. Please redo the references section.

1. Many of the references appear to be missing journals, and then the title ends up being formatted like the journal. Examples include Cunningham et al (1976), Efron and Tibshirani (1996).

   We thank the reviewer for catching these errors. They have been corrected and all references have been reviewed.

2. The Debus et al (2018) citation has no information other than author, title, and year, and the fact that it was accepted in some journal. This is a problem because section 2.3 cites Debus et al (2018) heavily. I tried to find it by searching online but could not.

   We apologize for the error. At the time, the manuscript was only under review, but now is accepted and available online http://dx.doi.org/10.1177/0003702818804574.

3. Book and grey literature references are incorrect, e.g. Mahalanobis, P "On the generalised distance in statistics", Tibshirani (2014) References:

   - Malm, W. C., and Hand, J. L.: An examination of the physical and optical properties of aerosols collected in the IMPROVE program, Atmospheric Environment, 41, 3407-3427, https://doi.org/10.1016/j.atmosenv.2006.12.012, 2007.

   - Solomon, P. A., Crumpler, D., Flanagan, J. B., Jayanty, R. K. M., Rickman, E. E., and McDade, C. E.: U.S. National PM2.5 Chemical Speciation Monitoring Network- CSN and IMPROVE: Description of networks, Journal of the Air & Waste Management Association, 64, 1410-1438, 10.1080/10962247.2014.956904, 2014.

   - Weakley, A. T., Takahama, S., and Dillner, A. M.: Ambient aerosol composition by infrared spectroscopy and partial least-squares in the chemical speciation network: Organic carbon with functional group identification, Aerosol Science and Technology, 50, 1096-1114, 10.1080/02786826.2016.1217389, 2016.

   We thank the reviewer for the additional references for the monitoring networks — they have now been included in the paper.

Additional comments from Quick Review Report

This paper appears to be quite good and should be given further review. My initial thought was to wonder if this was really a 'review article' - or just a very (high quality) extensive paper which describes the methods better. For example, the 'number of samples' section (3.5.2) cites only one paper, which was written by one of the co-authors. However, in the end, I thought that it has the potential to be the kind of paper I would want to read if I did want an overview of the steps taken by the authors, so I think it is OK as a review article.

In addition to a detailed description of methods, we believe it is a review in the sense that the paper synthesizes findings from past work on the topic of calibration with FT-IR spectra with collocated measurements. (The review covers mostly by the authors' work as there has not been much work published in this regard.) On the topic of carbon estimation, for instance, spectral preparation and model selection have been treated

differently in different works, and so this paper provides a broader perspective in which commonalities (and differences) in the approaches are discussed.

A few minor points:

1. The table of contents should be removed. I surveyed several other review articles in AMT and did not find any table of contents. However, this is a long article, perhaps the authors could consider adding it to a supplement.

   We have moved this to the appendix.

2. The authors should add an appendix which has definitions of all acronyms. It would help for readability in the interactive discussion. Again, this is a long article, it can be hard to find where an acronym is defined.

   We have added Table B1 which covers acronyms used in multiple sections.

3. Similarly, I recommend also defining site locations in the appendix (e.g. BYIS).

   We have included site acronyms that are used in multiple sections in Table B1.

4. Data Availability: The paper states that the data from IMPROVE and STN will be made available - does that apply to the FTIR spectra as well?

   The FT-IR spectra will be made publicly available, but at current time can be obtained by request from the authors.

Technical corrections:
'Atmosphere' is misspelled on line 5 of the abstract corrected
There is only TEXT for the copyright statement (Page 3, line 4) This will be revised upon publication.